



# Signature and sensitivity-based comparison of conceptual and process oriented models, GR4H, MARINE and SMASH, on French Mediterranean flash floods

Haruna Abubakar[1], Garambois Pierre-Andre[2], Roux Helene[3], Javelle Pierre[2], and Jay-Allemand Maxime[4]

[1]Univ. Grenoble Alpes, IGE, 38000 Grenoble, France
[2]INRAE, Aix Marseille Université, RECOVER, 3275 Route de Cézanne, Aix-en-Provence, 13182, France
[3]Institut de Mécanique des Fluides de Toulouse (IMFT), Université de Toulouse, CNRS - Toulouse, France
[4]Hydris Hydrologie corp., Montpellier, France

**Correspondence:** Pierre-André Garambois (pierre-andre.garambois@inrae.fr)

**Abstract.** The improvement of flood forecast ability of models is a key issue in hydrology, particularly in Mediterranean catchments that are subjected to strong convective events. This contribution compared models of different complexities, lumped GR4H, continuous SMASH and process-oriented MARINE. The objective was to understand how they simulate catchment's hydrological behavior, the differences in terms of their simulated discharge, the soil moisture, and how these can help to

improve the relevance of the models. The study was applied on two Mediterranean catchments in the South of France. The methodology involved global sensitivity analysis, investigations of the response surface, calibration and validation, signature comparison at event scale, and comparison of soil moisture simulated with respect to the outputs of the surface model, SIM. The results revealed contrasted and catchment specific parameter sensitivity to the same efficiency measure and equifinality issues are highlighted via response surface plots. Higher sensitivity is found for all models to transfer parameters on the Gardon and

for production parameters on the Ardeche. The exchange parameter controlling a non conservative flow component of GR4H is found to be sensitive. All models had good calibration efficiencies, with MARINE having the highest, and GR4H being more robust in validation. At the event scale, indices of discharge showed that, the event based MARINE was better in reproducing the peak and its timing. It is followed by SMASH, while GR4H was the least in this aspect. SMASH performed relatively better in the volume of water exported and is followed by GR4H. Regarding the soil moisture simulated by the three models

and using the outputs of the operational surface model SIM as the benchmark, MARINE emerged as the most accurate in terms of both the dynamics and the amplitude. GR4H followed closely while SMASH was the least in comparison. This study paves the way for extended model hypothesis and calibration-regionalization methods testing and intercomparison in the light of multi-sourced signatures in order to assess/discriminate internal model behaviours. It highlights in particular the need for future investigations on combinations of vertical and lateral flow components, including groundwater exchanges, in distributed

hydrological models along with new optimization methods for optimally exploiting, at the regional scale, multi-source datasets composed of both physiographic data and hydrological signatures.



# 1 Introduction

Performing accurate flood forecasts in terms of location, magnitude and timing of runoff and flooding remains a key challenge especially for intense convective rainfall events affecting Mediterranean areas. This need is particularly acute given
the potential intensification of the frequency of extreme precipitations in this region (e.g. Pujol et al. (2007); Tramblay et al. (2013); Tramblay and Somot (2018)), in which Mediterranean climate is characterized by a significant variability with warm and dry summers and heavy rainfall events in autumn (Drobinski et al., 2014). Nevertheless, given the complexity of the hydro-meteorological processes involved and their heterogeneous and limited observability, flash floods hydrological modeling remains a hard task and internal fluxes are generally tinged with large uncertainties.

The "resolution–complexity continuum" (Clark et al., 2017) has been investigated over the past 5 decades by many studies with various modeling approaches, ranging from point-scale processes numerically integrated at larger scales (e.g. catchment) to spatially lumped representation of the system response (Hrachowitz and Clark, 2017). Among the variety of existing hydrological models, and hypothesis they rely on, their components generally describe water storage and transfer (e.g. Fenicia et al. (2011)) via various combinations and parameterizations of vertical and lateral storage-flux operators. All hydrological
models are to some degree conceptual and due to limitations and uncertainties in their structure, parameters representativity, data availability, and even initial and boundary conditions, calibration/learning is generally required.

Whatever their status and complexity, hydrological models are most often calibrated and validated using integrative discharge time series at the outlet of a catchment (Sebben et al., 2013). However, multiple models configurations and associated parameters can lead to similar value of discharge (unicity problem so-called equifinality in hydrology (Beven, 2001)). Whereas a model
can be capable of reproducing the system response (e.g. discharge) it has been trained for, it can fail in reproducing meaningful system-internal dynamics and patterns (Hrachowitz and Clark, 2017), thus providing right answers for wrong reasons (Kirchner, 2006). Then arises the problem of better calibrating/validating hydrological models, and in particular distributed models, which makes it possible to take into account the spatial variabilities in the properties of the basins and atmospheric signals, to simulate spatialized hydrological quantities, but are confronted to the problem of equifinality and over-parameterizations (see
discussion in Jay-Allemand et al. (2020) in a flash flood context with spatially distributed calibration of SMASH model).

Recent works have investigated the effect of various modeling strategies on the performance at modeling discharge in some flash flood cases. Lobligeois et al. (2014) in a study on 181 catchments in France to check the effect of higher rainfall and conceptual model resolution on streamflow simulation have shown that semi distributed approach based on GR4 model (Mathevet, 2005) has performed better on Cévennes and Mediterranean regions where the rainfall spatial variability is very high. Boithias
et al. (2017) compared the performance of the distributed event-based MARINE model and the lumped continuous SWAT model in flash flood modeling on a French Mediterranean catchment, and found that while MARINE model simulated the peak and timing better, SWAT model was better at simulating the recession discharge and the exported water volume. Jay-Allemand (2020) proposed a variational (assimilation) algorithm and showed its potential for spatially distributed calibration of SMASH model parameters on a flash flood prone catchment.



In addition to river discharge, surface runoff controlled by soil infiltration rates, is also a key factor for flash floods simulation (Berthet et al., 2009; Douinot et al., 2018; Vincendon et al., 2010). Reaching coherent representation of state-fluxes variabilities both at the outlet and within catchments remains a challenge in spatially distributed modeling which could be moved ahead using the information from hydrological signatures (see review in McMillan (2020) and references in Bouaziz et al. (2021)) in combination with sensitivity analysis (Horner, 2020). Information selection and distributed model constrain can benefit

from sensitivity analysis as done with the MARINE model on flash flood Mediterranean catchments by Roux et al. (2011) or Garambois et al. (2013), guiding the design of regionalization methods accounting for bedrock types among other descriptors (Garambois et al., 2015). In the case of Mediterranean flash floods, Eeckman et al. (2020) recently assessed multi-hypothesis modeling of subsurface flows (Douinot et al., 2018) with MARINE using multi-sourced local and gridded soil saturation signatures.

The present study is aimed at understanding how models of varying complexity, namely simple conceptual, lumped or distributed, and process oriented distributed hydrological models, enable to simulate flash flood prone catchment behavior: what are the differences between the simulated dynamics, of both outlet discharge and internal states, and how this understanding can be used to improve the relevance of the models? To address these questions a methodology is designed based on global parametric sensitivity analysis, calibration-validation, analysis of response surfaces, performances and simulated signatures. We

consider two flash flood prone catchments in the South of France and three models of increasing complexity for hydrological modeling analysis: lumped conceptual model GR4H (Génie Rural) (Mathevet, 2005), spatially distributed conceptual models SMASH (Spatially-distributed Modeling and ASsimilation for Hydrology) (Jay-Allemand et al., 2020) with a Green and Ampt infiltration component, process oriented distributed model MARINE (Modélisation de l'Anticipation du Ruissellement et des Inondations pour des évéNements Extremes) (Roux et al., 2011). The parameters sensitivity and identifiability is investigated

for each model followed by a split sample calibration-validation with global performance analysis in time. Signatures analysis is performed at flood event scale considering simulated discharge features and soil moisture patterns from the operational surface model SIM (Habets et al., 2008). Finally a global sensitivity analysis is performed for event soil storage capacity evolution which is a critical quantity involved in flood flows genesis. It is related to both models parameters and simulated hydrological processes.

This paper is organized as follows: section 1 introduces the objectives and scope of the study. Section 2 details the models, tools and data. Section 3 details the methodology. Results are analyzed and discussed in section 4 and conclusions and perspectives are presented in section 5.

## 2 Models, Tools and Data

The approach is based on three hydrological models of increasing complexity that are presented here along with their calibration

methods. The regional sensitivity analysis method is detailed and next the study area and data.





## 2.1 Hydrological Models

Three hydrological models of varying complexities are used for this study: GR4H (lumped and conceptual), SMASH (spatially-distributed and conceptual), MARINE (process-oriented and spatially distributed). GR4H and SMASH are continuous models whereas MARINE is event based but its state is initialized with the outputs of the SIM operational surface model. All models

represent a limited number of hydrological processes and some of their flow operators share similarities as analyzed later in this study. Note that GR4H is the only model in this study with a "non conservative" flow operator. This section presents the formulation of all the models and their flow operators are detailed in appendix A

This section also presents the calibration algorithm of each model, used to optimize their parameters in order to reduce the discrepancy between simulated and observed discharges at a catchment outlet. The objective function used for calibration is

the classical NSE efficiency (given in section 3.4) that is adequate for the present flood modeling context. For all models, considering $J = 1 - NSE$, the parameter calibration inverse problem reads:

$$\theta^* = \arg\min_{\theta} J(\theta)$$

where the cost function $J$ depends on the sought model parameters $\theta$ through hydrological model response. For each model, bound constrains are applied on the sought parameters using the same ranges as in sensitivity analysis (cf. section 3).

We consider a 2D-spatial domain $\Omega$ (catchment) covered by a regular rectangular grid of resolution $\Delta x$. The unique constrain applied to this lattice is that a unique point has the highest drainage area, that is catchment outlet, given flow directions. The time is denoted $t > 0$. The spatio-temporal rainfall and evaporation fields are respectively $P$ and $E$, stepwise approximations over time steps $\Delta t$ are assumed.

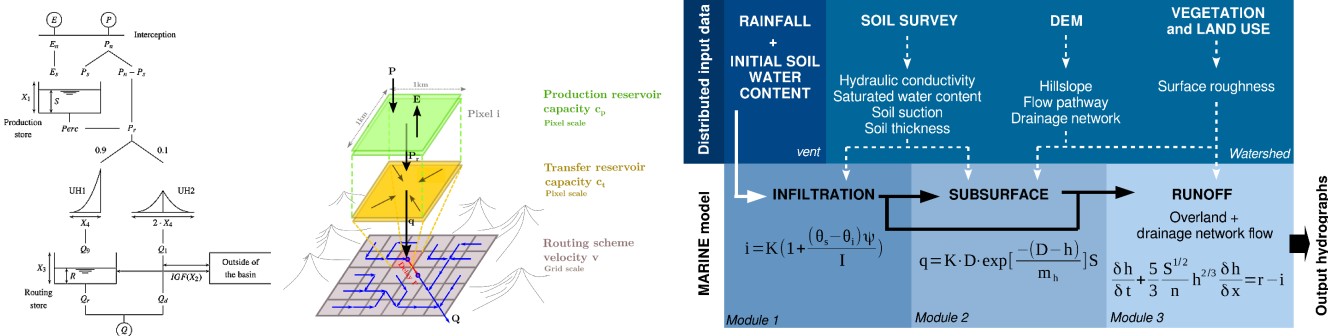

**Figure 1.** Conceptual representation of the three models used**:** (left) GR4 model lumped model structure from Lemoine et al. (2007), (middle) SMASH model structure with 3 flow operators from Jay Allemand et al. (2019), (right) MARINE model structure from Douinot et al. (2018).

### 2.1.1 GR4H model

The GR4H model (Mathevet, 2005) is a lumped continuous model that runs at the hourly time step and based on the GR4J model formulation of Perrin et al. (2003). This model has been used in many studies such as flash flood modeling in four





tropical mountainous watersheds in New Caledonia (Desclaux et al., 2018), for testing the transferability of the GR4H model parameters for extreme events in Mediterranean island of Cyprus (Caligiuri et al., 2019) or for comparison of two satellite estimated precipitation products in hydrological simulations in Rimac Basin, Peru (Astorayme and Felipe, 2019), among many others.

The partition of the input neutralized rain $P_n$ (cf. A1.1) is done between an infiltration part $P_s$ filling the production reservoir and an effective rainfall $P_r = P_n - P_s$ inflowing the transfer components. The production function is the classical GR production function (Edijatno and Michel, 1989), which is a soil moisture accounting model recalled in section A1 that gives the effective rainfall $P_r$ which is then splitted into two different kinds of flows. The splitting of the effective rainfall takes into account quick and slow flow components. The 10% of the effective rainfall $P_r$ resulting from the excess of the production and the percolation is routed linearly using a unit hydrograph UH2 of time base $2x_4$, the remaining 90% is initially routed using UH1 of time base $x_4$ and then using a non linear routing store of reference capacity $x_3$. The ordinates of the UH are derived from their respective S hydrographs which also are functions of $x_4$. A groundwater flow exchange term $F$ from the reservoir which depends on both the actual level in the routing store $R$, the reference level of the non-linear routing store $x_3$ and a water exchange coefficient $x_2$ is taken into account in both flow components. Finally, the total stream flow $Q$ is obtained as the sum of the resulting flows from the routing reservoir $Q_r$ and the output of the UH2 $Q_d$.

**Table 1.** Description of the GR4H model parameters and range used for sensitivity analysis

| Parameter | Description | Unit | Range |
|:---:|:---:|:---:|:---:|
| $x_1$ | production storage capacity | mm | 1 - 1500 |
| $x_2$ | groundwater exchange coefficient | mm | -10 - 10 |
| $x_3$ | max. capacity of the routing store | mm | 0 - 500 |
| $x_4$ | time base of the unit hydrograph UH1 | hours | 0 - 10 |

For the GR4H model, four parameters $x_1$, $x_2$, $x_3$, $x_4$ are optimized (see Table 1). The calibration is done using the "Michel calibration algorithm", which starts with random starting points in the parameter space and optimum search is performed with a simple descent method. The choice of the initial conditions of the model for example the production store level $S$ greatly affects the result for the first year of simulation - warm up.

### 2.1.2 SMASH model

SMASH (Spatially-distributed Modeling and ASsimilation for Hydrology) is a computational software framework dedicated to spatially distributed continuous hydrological modeling including variational data assimilation (Jay-Allemand et al., 2020). We use the 3 components model (production, transfer, routing) from Jay-Allemand et al. (2020). For a given pixel $i$ of coordinates $x \in \Omega$ two reservoirs $\mathcal{P}$ and $\mathcal{T}$, of capacity $c_p$ and $c_{tr}$, are considered for simulating respectively the production of runoff and its transfer within a cell. Their stages are respectively denoted $h_p$ and $h_{tr}$. The runoff amount is then routed between pixels.





The spatial resolution is set to $\Delta x = 1\text{km}^2$ and the simulation time step is set at $\Delta t = 1h$ correspondingly to the space-time resolution of rainfall data.

The partition of the input neutralized rain $P_n$ (cf. A1.1) between an infiltration part $P_s$ filling the production reservoir and an

effective rainfall $P_r = P_n - P_s$ filling the transfer reservoir is done with a production operator. In this study, a Green and Ampt infiltration model (eq. A3) enabling to simulate ponding when rainfall intensity exceeds infiltration rate, has been implemented in the model (differentiation of source code with TAPENADE, classical tests performed - not shown). The production reservoir is then emptied from an actual evaporation $E_p$ calculated with a "GR" evaporation operator (eq. A2).

The effective rainfall after production is transferred within a pixel through a conceptual reservoir of maximum capacity $c_{tr}$

(eq. A4), while routing is done with a linear unit Gaussian hydrograph whose delay $\tau_i$ from node $i-1$ to node $i$ is controlled by the routing velocity $v$ and the distance $d_i$ between the cells. The model formulations are described in Appendix A1.

The complexity of a hydrological modeling approach also lies in its spatialization. The variational algorithm presented in Jay-Allemand et al. (2020) and enabling the calibration of spatially distributed model parameters, that is high dimensional optimization problems, under various constrains is performed. This variational calibration algorithm starts from a spatially

uniform prior guess on the sought parameters (Jay-Allemand et al., 2020). This prior guess is obtained with a simple global calibration algorithm as in Jay-Allemand et al. (2020). The minimization of the cost function is then done using the LBFGS-B (Limited memory Broyden-Fletcher-Goldfarb-Shanno Bound-constrained) descent algorithm (Zhu et al., 1994) making use of the gradient of the cost function that is obtained from the adjoint model thanks to the Tapenade automatic differentiation engine (Hascoet and Pascual, 2013).

However, using only downstream discharge for calibration leads to well known controlability issues in spatially distributed hydrological modeling. A reduction of the control space is done by applying spatial masks derived from prior physiographic information to group the sought parameters by classes (Jay-Allemand, 2020). Note that the same reduction of the control space is used for the sensitivity analysis performed before calibrations in this study. For example, in the case of Gardon (543 $km^2$, hence 543 pixels of 1 $km^2$) instead of calibrating ($4 \times 543 = 2172$ parameters), a mask is used for each parameter. If the mask

for the routing parameter $v$ has only two classes (one for the drainage network and another for the hillslope), then only two $v$ parameters will be optimized (instead of 543 pixel values). A key task is to find relevant spatial information to define the mask for the parameters of a model that is conceptual (SMASH). In Jay-Allemand (2020), different masks have been proposed and tested. However for the present intercomparison study, similar physiographic maps used for the MARINE model are used to define the parameter masks for SMASH. They are summarized in the Table 2. Four free parameters $c_p, c_{tr}, v, k_s$ times their

respective number of classes defined by their masks (Table 3) are considered for calibration. Suction $Sf$ and porosity $Poros$ are not calibrated, based on previous sensitivity analysis of Green and Ampt model in a similar context (Garambois et al., 2015; Roux et al., 2011). While $Sf$ is defined using prior soil information (Table 2), $Poros$ is simply kept at a value of 1 (see A1.2). This calibration method from Jay-Allemand (2020) considering semi-distributed patterns of the SMASH model parameters is called "masked" calibration in the following. The use of physiographic maps to define spatial patterns is detailed after.





**Table 2.** Prior Information used to define parameter masks for SMASH parameters. The soil classes are defined from the soil texture using the Rawls and Brakensiek (1983) relations, from which the $k_s$, and $Sf$ are obtained. Only the first 4 (resp. three) parameters $c_p$, $c_{tr}$, $v$, $k_s$ (resp. $c_p$, $c_{tr}$, $v$) are calibrated as a result of the sensitivity analysis.

| Parameter | description | Prior information |
|:---:|:---:|:---:|
| $c_p$ | Production reservoir capacity | Map of soil thickness |
| $c_{tr}$ | Capacity of the transfer reservoir | Map of slope |
| $v$ | Routing velocity | Flow accumulation maps |
| $k_s$ | Saturated hydraulic conductivity | Map of the soil hydraulic conductivity from texture map |
| $Sf$ | Soil suction | Map of the suction from texture map |

### 2.1.3 MARINE model

MARINE is an event, physically based, parsimonious and fully distributed model designed for flash flood prediction based on the supposedly main hydrological processes involved in Mediterranean catchments. It is borne out of the need to address the peculiarities identified by Roux et al. (2011) in different models ranging from domain of applicability (floods), difficulty of accessing data for model calibration and inability of the present models to help improve the understanding of the hydrological processes that are specific to Mediterranean catchments. The processes of infiltration, subsurface runoff, overland flow and flow in drainage networks are represented while the processes of evaporation and deep percolation are considered not important at the event scale and therefore not represented.

MARINE being an event based model, the local infiltration function used is a typical event based model, accounting for the infiltration at the local scale and described by the Green and Ampt model (Equation A3).

The surface runoff is divided into overland flow and drainage flow, in both cases, the kinematic wave model is used assuming a 1-dimensional kinematic wave which is approximated with the Manning friction law while the subsurface flow is based on the Darcy's law. The model formulations are given in Appendix A1.

Input data are sourced from information of surface topology, soil survey, vegetation and land use, and the model is initialized using soil moisture outputs of the SIM model. Finally, the model requires only five parameters to be calibrated for the whole catchment; three correction coefficients applied to the distributed maps of saturated hydraulic conductivity $C_k$, the soil thickness $C_z$, and the soil lateral transmissions $C_{kss}$, the other two parameters include Manning-Strickler's friction coefficient for the river bed $K_{D1}$ and for the flood plain $K_{D2}$. These correction coefficients are applied during the calibration process such that the absolute values of the parameter in question is modified while the spatial pattern as sourced is preserved. The use of physiographic maps to define spatial patterns is detailed after.

The model has been used in several studies (e.g. Le Xuan et al. (2006); Garambois (2012); Garambois et al. (2013, 2014, 2015); Boithias et al. (2017); Douinot et al. (2018); Eeckman et al. (2020)). The spatial resolution is set to $\Delta x = 500m^2$ and the fixed simulation time step is set to $\Delta t = 6\ min$ (CFL check and automatic temporal sub-iterations if needed for kinematic wave resolution), i.e. finer than rainfall space-time resolution.





Calibration of the MARINE model is done by comparing the simulated and observed discharge with NSE as the objective
function. The optimization algorithm in the case of this model is based on a gradient-based descent algorithm BFGS (Broyden-
Fletcher-Goldfarb-Shanmo) from multiple starting points (Roux et al., 2011). The gradient is evaluated by finite differences.
The calibration involves estimating for a given event the values of the three correction coefficients applied to the distributed
maps of saturated hydraulic conductivity $C_k$, the soil thickness $C_z$, and the soil lateral transmissions $C_{kss}$. The other two
parameters include Manning-Strickler's friction coefficient for the river bed $K_{D1}$ and for the flood plain $K_{D2}$.

### 2.1.4 SIM Model

SIM, acronym for SAFRAN- ISBA -MODCOU (Habets et al., 2008), is an operational modeling chain that simulates both flow
of water and energy at the surface, as well as the flow of rivers and the major aquifers. It is forced by the atmospheric reanalysis
from SAFRAN, uses ISBA to simulate the exchange of water and energy between the soil and atmosphere; and MODCOU as
the hydrological model.

The two versions of the SIM model are used in the present study, that is SIM1 and SIM2. The first version SIM1 is used
simply for the initialization of the MARINE model as has been used by several authors (see Garambois (2012); Garambois
et al. (2013); Douinot et al. (2018); Eeckman et al. (2020) ), while the second version, SIM2 is used as the benchmark to
compare the simulated soil moisture outputs of the SMASH, GR4H, and the MARINE model.

The first version, SIM1, uses the force-restore version of ISBA, ISBA-3L (Noilhan and Mahfouf, 1996; Noilhan and Planton,
1989) in which the soil is discretized into three layers corresponding to surface, root and deep zone. SIM2 on the other hand uses
the diffusive version of ISBA, ISBA-DIF (Decharme et al., 2011), with a vertical soil column discretization into a maximum
of 14 layers. In the case of this study, the humidity of the root zone is considered as the sum of the humidities of the layers
between 10 $cm$ and 30 $cm$ deep.

The two outputs (SIM1 and SIM2) available for this study are at a daily time step (06 UTC) and a spatial resolution of 8 $km$
square grid.

## 2.2 Study Area and Data

### 2.2.1 Catchments

The study catchments (Gardon at Anduze and Ardeche at Vogue) are located in the Cevennes region, prone to flash flood and
are influenced by a Mediterranean climate. Data types and sources are described in the next paragraph.

The Ardeche catchment at Vogue drains an area of 622 km$^2$, and is exposed to intense precipitation events due to the
convection of humid sea air masses over the Cevennes mountain slopes (Eeckman et al., 2020). It presents a mixed geology,
with metamorphic rocks and schist on the upper part of the catchment, and sedimentary plains downstream. The land cover is
mainly mixed forest, natural grasslands and shrubs. The elevation varies between 1530 m at the upstream to 150 m downstream.
The depth of the soil in the catchment ranges between as low as 5 $cm$ to as deep as 50 $cm$ with an average depth of 28 $cm$.





The soil texture is mainly sandy-loam with silt deposits downstream. The mean saturated hydrological conductivity is around
8.6 $mm/hr$.

  The Gardon with its outlet at Anduze drains an area of 540 km². It is well gauged and has a Mediterranean climate with a lot
of intense rainfalls in the autumn and winter. It is characterized by the occurrence of flash floods and the highest rainfall rates
in autumn, while the summer is mostly hot and dry (see Roux et al. (2011)). The catchment geology is mainly dominated by
fractured metamorphic formation, classically the schistose, however there are some karstic zones around the junction of Saint
Jean and Mialet (Le Xuan et al., 2006). It has a highly marked topography consisting of high mountain peaks, narrow valleys
and steep hill slopes. The vegetation is dense and composed mainly of beech, chestnut trees, holm oaks and conifers (Moussa,
2010). The elevation varies between 129 m at Anduze to 1202 m at the highest point. The average slope of the basin is about
20%, but can be up to 50% at the upstream. The soil (made of by silty-clay loam and sandy loam) has a mean thickness of
around 28 $cm$ and a mean saturated hydraulic conductivity 5 $mm/hr$.

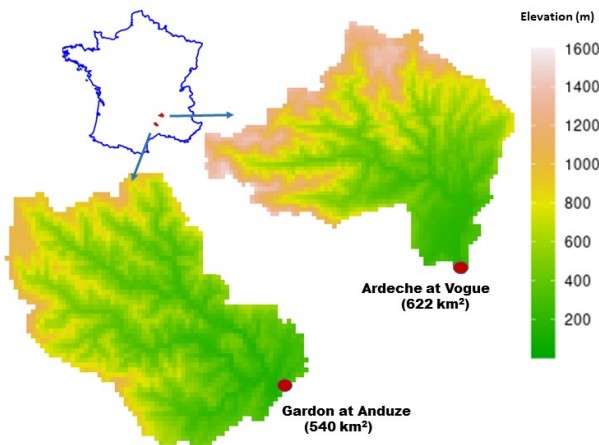

**Figure 2.** Map of the two study catchments, both located in the South of France. Top left: map of France showing the location of the two
catchments in red. Top right: Ardeche at Vogue, Bottom left: Gardon at Anduze. The area of both catchment are shown. On both catchments,
the position of the outlets is shown by the red circle. The legend represents the elevation in $m$, with a spatial resolution of $500\ m^2$, with
respect to mean sea level.

### 2.2.2 Data

To provide a fair assessment of the models, the same input of discharge, rainfall and for the specific case of the continuous
models (SMASH and GR4H), potential evapotranspiration (PET), are used. The hourly discharge have been extracted from the
HYDRO database of the french ministry in charge of environment while the rainfall data from the radar observation reanalysis
ANTILOPE J+1, which merges radar and insitu gauge observations, is provided by Météo-France. The interannual temperature
data is provided by the SAFRAN reanalysis and then used to calculate the potential evapotranspiration using the Oudin formula





Oudin et al. (2005). The rainfall and PET are at a spatial resolution of 1 $km^2$ square grid, and processed into hourly time step. Spatial averages of the rainfall and PET over the catchments are used as input for the lumped GR4H model. The soil thickness and texture maps are derived from the surveys provided by the INRA and BRGM. Soil classes and consequently the suction,

porosity and saturated conductivity are derived from the soil texture using the Rawls and Brakensiek (1983) relations. The vegetation and land use from the (2000 Corine Land Cover: Service de l'Observation et des statistiques) is used to derive the surface friction. This is exactly the same data used and sourced from Roux et al. (2011). The resulting maps are used as inputs for the MARINE model to provide physical operator parameters values, while they are used as mask inputs for the SMASH model in the calibration by classes (masked calibration) (Refer to Table 2 . As explained in section 2.1.4, the SIM1 soil moisture

outputs are used for the initialization of the MARINE model, while the SIM2 outputs are used for model inter-comparison.

## 3   Numerical Experiments Methodology

Sequel to the motivations and the objectives of the study as outlined in the introduction, the following numerical experiments are designed to help answer the questions raised. The first experiment is designed to investigate the global sensitivity of the three models. Experiments for the calibration and validation of the models within the study period are then followed, and finally

the methodology to compare the model performance at the event scale is described. The evaluation criteria used are also briefly described.

### 3.1   Regionalized Sensitivity Analysis

The sensitivity analysis of the model parameters is based on the regionalized sensitivity analysis (RSA) approach (see Appendix 3.1). The aim is to identify the parameters to which the model is most sensitive to. In this section, we describe how the sensitivity

analysis methodology is applied to each model.

**Table 3.** Description of SMASH parameters and ranges used for calibration and sensitivity analysis on the study catchments

| parameter | Description | Range | No of classes | |
|:---:|:---:|:---:|:---:|:---:|
| | | | Ardeche | Gardon |
| $c_p$ | Capacity of the production reservoir (mm) | 1 - 2000 | 4 | 124 |
| $k_s$ | Saturated hydraulic conductivity (mm/hr) | 0.1 - 20 | 12 | 12 |
| $c_{tr}$ | Capacity of the transfer reservoir (mm) | 1 -1000 | 5 | 5 |
| $v$ | Routing velocity (m/s) | 1/6 - 5 | 2 | 2 |

First, in the case of the GR4H model which is lumped, 10,000 simulation runs are done using randomly (uniform) generated parameters of the model within their range (see Table 1). The parameters are $x_1$, $x_2$, $x_3$ and $x_4$. For the current and subsequent experiments, a threshold of 0.7 NSE (eq. 1) is used for the classification of the runs in to behavorial (runs with NSE $\geq$ 0.7) and non-behavioral (NSE $<$ 0.7) groups. As noted by Beven (2001), the KS test can be very sensitive to small differences, and





will thus report significant differences between the two classes. Hence, the magnitude of the KS statistics $D$, representing the maximum difference between the cdf of the two classes is used to rank the parameters based on their sensitivity.

Secondly, in the case of the SMASH model, which is a fully distributed model at a spatial grid of $1 \text{ km}^2$, classical reduction of the high dimensional control space is adopted using physiographic masks (as described in section 2.1.2). The sensitivity analysis is then performed on the resulting parameters. The two reduction approaches are described below:

1. In the first instant, the parameters of the model are taken as being spatially uniform, and therefore, the RSA is done assuming one parameter set at a time for the whole catchment considered. The four parameters are: $c_p$, $c_{tr}$, $v$, and $k_s$. The sampling of the parameters within their ranges specified in Table 3 and assuming uniform distribution is done for 10,000 Montecarlo simulation runs. The classical RSA described in section B is followed. The same threshold of $NSE = 0.7$ is used for the classification of the runs into behavioral and non-behavioral groups. The KS test statistics is 270    then calculated to rank the parameters' sensitivity.

2. In the second approach, the dimension reduction is through the use of parameter regionalization approach, defined by prior information on the parameters spatial distribution from basin predictors. The predictors include soil texture, thickness, land cover and topography. Pedotransfer functions (eg. Rawl and Brakensiek) are then used to relate the model parameters and the basin predictors, and then an upscaling operator (arithmetic mean) is used to upscale from the 275    predictor scale to the modeling scale ($1 \text{ km}^2$). The soil thickness map is used for the production reservoir capacity $c_p$, the map of saturated conductivity for the Green and Ampt saturated conductivity $k_s$, the map of the flow accumulation for the routing parameter $v$ and the map of the slopes for the transfer parameter $c_{tr}$. Afterwards, for each parameter, uniform random values within the specified range (see Table 3) is generated for each class. For example, if for a catchment, the classes of the soil thickness are five, then five random values are generated, each one for a class, yielding a total 280    of five generated values per run for the $c_p$ parameter (the soil thickness being the predictor of $c_p$). The five values will be mapped to their grid location in the parameter map. 10,000 simulations are then run and the corresponding NSE is calculated. In presenting the scatter plots (for identifiability investigations), average of the mask (raster values) weighted by the class percentage is shown since it is not possible to show the whole raster on the plot.

Lastly, the sensitivity of the five MARINE model parameters (see Table 4) is investigated using the NSE criteria. The 285    eight events for Gardon and seven events for Ardeche are used for the analysis (see Table 5). 5000 runs were conducted as previously done by Roux et al. (2011) and Boithias et al. (2017) for each event. The same threshold value of 0.7 is used and a classification is done for each event. Unlike the case of Boithias et al. (2017) and the references reported therein, where the result of the sensitivity analysis was used to choose calibration/validation events, our methodology here is basically to investigate the parameter sensitivity. The method for the choice of the calibration/validation events is described in section 3.2.

290    **3.2   Calibration and Validation**

For each of the three hydrological models GR4H, SMASH and MARINE, parameters calibration is performed with their dedicated methods presented in section 2.1. Those methods enable adequate calibrations for each model as shown after. In





**Table 4.** Range of parameters for the sensitivity analysis of MARINE model

| Parameter | Description | Gardon |
|:---:|:---:|:---:|
| $C_k$ | Correction coefficient of the hydraulic conductivities | 0.1-10 |
| $C_z$ | Correction coefficient of the soil thicknesses | 0.1-10 |
| $C_{kss}$ | Correction coefficient of the soil lateral transmissivities | 100-10000 |
| $K_{D1}$ | Strickler's friction coefficient of the river bed | 1-30 |
| $K_{D2}$ | Strickler's friction coefficient of the flood plain | 1-20 |

order to perform fair comparisons, considering comparable amount of hydrological information learnt by the models in the calibration phase, the calibration and validation is done using a split-sample test procedure by dividing the data into two (Klemeš, 1986). Time series of 13 years at hourly time step is considered and the sub-periods of 7 years each for the calibration and validation are defined as period 1 (1st August 2006 to 1st August 2013) while period 2 (1st August 2012 to 1st August 2019). Calibration is done first using period 1 and then validation on period 2, the reverse is then done in which period 2 is taken as calibration period while period 1 is taken for validation. For each calibration period, 1 year is used as the warm up period to initialize the model which is adequate for hydrological models as reported by Kim et al. (2018). In the case of MARINE, the events (see Table 5) are classified into two periods (similar to the continuous models) and a multi-event calibration and cross validation is done. Similar multi-events calibration of MARINE has been carried out by Garambois et al. (2013). For all the calibrations, we used the NSE as the objective function.

### 3.3 Comparison of models at event scale

These experiments are designed to compare the three models at flash flood modeling. The inter-comparison involves the assessment of key indices of peak flow estimation as well as its timing and the internal fluxes simulated. In addition to the NSE criteria, the percentage peak difference (PPD), peak delay (PD) as well as synchronous percentage of the peak discharge (SPPD) introduced in section 3.4 are used.

The "soil moisture" simulated by the model are also compared with outputs of the SIM model. While SIM outputs are at a spatial resolution of 8 $km$, those of SMASH and MARINE are at 1 and 0.5 $km$ respectively. In the case of GR4 that is lumped, the resolution is at the scale of the catchment size. We compare the soil moisture by looking at the temporal evolution of the spatially averaged outputs of each model, and how close it is to those of the SIM2 outputs, which in our case is the reference benchmark.

Specific flood events of return period higher than 2 years are chosen within the period of 13 years (2006-2019) for both catchments, They are given in Table 5. These selected events provided distinct characteristics in terms of the flood peak magnitudes, the volume of water exported, the number of peaks, the gradients of the rising and falling limbs as well as the spatial and temporal patterns of the underlying precipitation events.

In order to provide a fair comparison, the same rainfall forcing and discharge data are used for all the models.





**Table 5.** Selected flood events for comparison of the model performance at event scale.

| Gardon | Season | Duration (days) | $Q_{obs}^{peak}$ (m³/s) | Vol (×$10^6$ m³) |
|---|---|---|---|---|
| Ev_31_10_2008 | Autumn | 4 | 1011 | 57.1 |
| Ev_02_11_2011 | Autumn | 6 | 1026 | 127.4 |
| Ev_17_09_2014 | Autumn | 5 | 1012 | 44.4 |
| Ev_09_10_2014 | Autumn | 7 | 1146 | 78.6 |
| Ev_11_09_2015 | Autumn | 2 | 980 | 29.6 |
| Ev_27_10_2015 | Autumn | 2 | 1356 | 33.4 |
| Ev_22_11_2018 | Autumn | 2 | 655 | 38.4 |
| Ev_08_11_2018 | Autumn | 2 | 809 | 27.6 |
| **Ardeche** | **Season** | **Duration (days)** | $Q_{obs}^{peak}$ (m³/s) | Vol (×$10^6$ m³) |
| Ev_2008_10_19 | Autumn | 5 | 954 | 68.8 |
| Ev_2010_05_11 | Spring | 2 | 420 | 18.3 |
| Ev_2010_09_06 | Autumn | 2 | 1272 | 29.8 |
| Ev_2011_11_02 | Autumn | 6 | 867 | 157.1 |
| Ev_2014_09_18 | Autumn | 3 | 1524 | 77.7 |
| Ev_2014_11_14 | Autumn | 2 | 1194 | 61.5 |
| Ev_2019_04_23 | Spring | 6 | 514 | 56.7 |

## 3.4 Performance Evaluation Criteria

In the course of all the calibration and the validation of the hydrological models used, the objective function used for the
calibration is the widely used Nash and Sutcliffe efficiency criterion: which puts more weights on the high flows than on low
flows, and is adapted to our objective of assessing the ability of the model to simulate flash floods.

$$NSE = 1 - \frac{\sum_{i=1}^{T}\left(Q_{s(i)} - Q_{o(i)}\right)^2}{\sum_{i=1}^{T}\left(Q_{o(i)} - \bar{Q}_o\right)^2} \tag{1}$$

where $\bar{Q}_o$ is the mean of observed discharges, $Q_{s(i)}$ and $Q_{o(i)}$ are simulated and observed discharges at time step $i$ respectively.

In the case of inter-model performance evaluation between the SMASH, GR4H and MARINE at event scale, other criteria
are used. They include:

– The Kling-Gupta Efficiency (KGE) (Gupta et al., 2009) which provides an alternative to the NSE and gives balance to
the correlation, flow variability and water balance.

$$KGE = 1 - \sqrt{(r-1)^2 + (\beta-1)^2 + (\alpha-1)^2} \tag{2}$$





$r = \frac{cov(Q_o, Q_s)}{\sigma_o^2 \sigma_s^2}$, the Pearson correlation coefficient, evaluates the error in shape and timing between observed ($Q_o$) and simulated ($Q_s$) flows, $cov$ is the co-variance between observation and simulation and $\sigma$ is the standard deviation, $\beta = \frac{\mu_s}{\mu_o}$, evaluates the bias between observed and simulated flows where $\mu$ is the mean. $\alpha = \frac{\sigma_s}{\sigma_o}$, the ratio between the simulated and observed standard deviations, evaluates the flow variability error.

- Percentage Peak Difference: This criteria is given as $PPD = \frac{Q_{p;sim}}{Q_{p;obs}}$ and is mainly to judge the percentage of the ob-
served peak predicted by the model, the duo must not coincide in time of occurrence.

- Peak Delay (PD): Given as $t_{p;sim} - t_{p;obs}$ and simply computes the difference in time or delay between the simulated and observed peak

- A more rigorous criteria in terms of safety is the synchronous percentage of the peak discharge (SPPD) that accounts for the ratio of the estimated discharge and observed discharge at the time of the observed peak discharge. It has been used
first by Artigue et al. (2012) and then subsequently by Jay-Allemand et al. (2020) and can be written as $\frac{Q_{sim}}{Q_{p;obs}}$

Finally we also use as a metric, the runoff coefficient (CR).

## 4 Results and Discussion

The results obtained after conducting the numerical experiments described in section 3 are presented here, along side relevant discussions.

The calibration and validation efficiencies as well as the event signatures are also presented and discussed. Finally, the comparison of the simulated soil moisture, as compared to the gridded outputs of SIM model are presented and discussed.

### 4.1 Sensitivity Analysis

The results obtained from the regionalized sensitivity analysis of the three models are presented in this section.

**Table 6.** Sensitivity ranks of the SMASH model parameters (left) and GR4 (right) computed according to the Kolmogorov-Smirnov test statistics, D, accounting for the maximum distance between the behavioral and non-behavioral distributions. (1 is the most sensitive, 4 is the least sensitive). In the case of SMASH, the result obtained through dimension reduction using spatially uniform and masked parameters are shown

| Catchment | Mode | $c_p$ | $c_{tr}$ | $v$ | $k_s$ |
|-----------|------|-------|----------|-----|-------|
| **Gardon** | Uniform | 3 | 1 | 2 | 4 |
| | Masked | 3 | 1 | 4 | 2 |
| **Ardeche** | Uniform | 2 | 3 | 1 | 4 |
| | Masked | 1 | 2 | 4 | 3 |

| Catchment | $x_1$ | $x_2$ | $x_3$ | $x_4$ |
|-----------|-------|-------|-------|-------|
| **Gardon** | 3 | 1 | 2 | 4 |
| **Ardeche** | 2 | 1 | 3 | 4 |





**Table 7.** Sensitivity ranks of the MARINE model parameters computed according to the Kolmogorov-Smirnov test statistics, D, accounting for the maximum distance between the behavioral and non-behavioral distributions. (1 is the most sensitive, 5 is the least sensitive)

| Gardon | $C_Z$ | $C_k$ | $C_{kss}$ | $K_{D1}$ | $K_{D2}$ |
|---|---|---|---|---|---|
| Ev_10_11_2008 | 2 | 3 | 1 | 5 | 4 |
| Ev_01_11_2011 | 1 | 3 | 2 | 4 | 5 |
| Ev_16_09_2014 | 2 | 3 | 1 | 5 | 4 |
| Ev_09_10_2014 | 4 | 3 | 1 | 2 | 5 |
| Ev_10_09_2015 | 2 | 4 | 1 | 3 | 5 |
| Ev_27_10_2015 | 2 | 3 | 1 | 5 | 4 |
| Ev_22_11_2018 | 2 | 4 | 1 | 3 | 5 |
| Ev_08_11_2018 | 2 | 3 | 1 | 5 | 4 |
| **Average** | 2.1 | 3.3 | 1.1 | 4.0 | 4.5 |

| Ardeche | $C_Z$ | $C_k$ | $C_{kss}$ | $K_{D1}$ | $K_{D2}$ |
|---|---|---|---|---|---|
| Ev_2008_10_19 | 3 | 1 | 2 | 5 | 4 |
| Ev_2010_05_11 | 2 | 1 | 5 | 3 | 4 |
| Ev_2010_09_06 | 2 | 1 | 4 | 5 | 3 |
| Ev_2011_11_02 | 1 | 4 | 4 | 3 | 2 |
| Ev_2014_09_18 | 2 | 1 | 4 | 3 | 5 |
| Ev_2014_11_14 | 4 | 2 | 5 | 3 | 1 |
| Ev_2019_04_23 | 3 | 1 | 2 | 4 | 5 |
| **Average** | 2.4 | 1.6 | 3.7 | 3.7 | 3.4 |

### 4.1.1 SMASH (uniform and masked)

**Uniform Parameters**

Figure 3 gives the results of the sensitivity analysis under spatially uniform parameter sets. In the case of the Gardon catchment, the scatter plot (first row) shows clear identifiability for the transfer parameter $c_{tr}$. The two production parameters $c_p$ and $k_s$ shows the least identifiability, while the routing parameter $v$ shows exclusive poor performance for small values. Under our tested methodology, peaky scatter plots for a parameter indicates a good identifiability. The scatter plots in the case of the Ardeche catchment shows a drop in performance for values of $c_p$ higher than 1200, below this value, both good and poor performances can be obtained. In the case of the $k_s$ parameter, the scatter plot shows clear non-identifiability due to clear randomness throughout the parameter range. The transfer parameter $c_{tr}$ appears to be peaky for this catchment also. Finally, similar to Gardon, the routing parameter $v$ shows significant drop in performance for small values.

The cumulative distribution of the behavioral and non-behavioral classes (second row) are based on the NSE threshold of 0.7. In the case of the Gardon catchment, $c_p$ exhibits flat slope for small ($< 125$) and high ($> 1750$) values with near uniform distribution in between, while the distribution of the non-behavorial classes is uniform showing that poor NSE can be obtained throughout the parameter range. In the case of the $c_{tr}$ parameters, which is also the most sensitive, the slope is non-zero only within very small range (between 200 and 400), outside this range, all realizations are poor. Relatively flat slope is observed within this range for the non-behavioral realizations confirming the absence of poor realizations within the range. The KS statistics $D$ is largest for $c_{tr}$ confirming that it is the most sensitive. For the case of Ardeche, although the scatter plot shows that $c_{tr}$ is most identifiable due to its peakedness, the test statistics shows $v$ to be the most sensitive and closely followed by $c_p$. However, $k_s$ still remains the least sensitive.





The transfer parameter observed to be the most sensitive has to do with the fact that the performance measure used is the NSE which gives more weight to high values. In the SMASH model, $c_{tr}$ controls the amount of the effective rainfall that is
transferred for routing and thus affects the magnitude and timing of the peak flows.

1. Scatter plots

2. Cumulative distribution of the behavorial and non behavorial classes

**Figure 3.** RSA of the four SMASH spatially uniform parameters on the two study catchments (left column: Gardon, right column: Ardeche). For each catchment, the first row shows the scatter plot of the NSE efficiency and the second row, the NSE cumulative distribution of the behavioral and non behavioral classes indicating the Kolmogorov-Smirnov statistics $D$.





**Masked Parameters**

The second RSA investigated is under the reduction of the control space through the use of masks due to high dimensionality resulting from the fully distributed nature of the model. As described in the methodology under section 3.1, the results are shown for the average of the mask (raster values) weighted by the class percentage as it is not possible to show the whole

raster on the plots. The scatter plots and the cdf shown in the first and second rows of Figure 4 respectively have the same interpretation as discussed and presented in the preceding paragraph 4.1.1. Peaky plots indicates identifiability and the larger the difference between the behavioral and non-behavioral classes, the more sensitive the parameter is. However, in this case there are some differences in the sensitivity of the model parameters. In the case of the Gardon catchment, $c_{tr}$ remained the most sensitive, but the routing parameter became the least sensitive. For Ardeche, $v$ which hitherto was the most sensitive

according to the KS test under uniform configuration, became the least sensitive, parameter while $c_p$ is the most sensitive. The differences between behavioural and non-behavioural distributions are less pronounced than with the uniform strategy, possibly because the results shown are the average of all the raster values.

### 4.1.2   GR4

In the case of the GR4H model, the RSA results for both catchments are presented in Figure 5. For both catchments, the time

base of the unit hydrograph $x_4$ is the least sensitive, while the ground water coefficient $x_2$ is the most sensitive. For the Gardon catchment specifically, the size of the production reservoir $x_1$ is less sensitive compared to the exchange coefficient $x_2$ and the routing store capacity $x_3$, whereas in the case of the Ardeche catchment, the sensitivity of $x_1$ is very close to that of $x_2$, the capacity of the routing store $x_3$ beeing the third most sensitive.

### 4.1.3   MARINE

The result of the sensitivity analysis of the MARINE model for both catchments is presented in Figure 6 and the summary of the parameter sensitivity ranks computed according to the KS test statistics $D$ is shown in Table 7. The ranking of the parameters is event dependent for each of the two catchments. In the case of the Gardon, the coefficient applied to the lateral subsurface flow, $C_{kss}$ emerged as the most sensitive for all the events except the Nov 2011 flood. It is then followed by the coefficient applied to the soil thickness, $C_z$. In other words, the three most sensitive parameters are related to the soil storage

capacity. The two Manning-Strickler's friction coefficients for the river bed $K_{D1}$ and the flood plain $K_{D2}$ emerged as the least sensitive in the ranking. In the case of the Ardeche catchment, different sensitivity ranks of the parameters are obtained. For this catchment, the correction coefficient $C_k$ of the hydraulic conductivity (infiltration) emerged as the most sensitive, which is then followed by $C_z$. Unlike the case of the Gardon, $C_{kss}$, along with $K_{D1}$ are the least sensitive.

The flood events in the Gardon are all autumn events, however the October 2014 flood appeared entirely different in terms of

the distribution of the behavioral realizations, because very few observations above the NSE threshold of 0.7 are obtained for this specific event. Ardeche on the other hand has two events occuring in spring, while the rest are autumnal. There is however no significant observable difference between the distributions of these events.





1. Scatter plots

2.Cumulative distribution of the behavorial and non behavorial classes

**Figure 4.** RSA of the four SMASH spatially masked parameters on the two study catchements (left column: Gardon, right column: Ardeche). For each catchment, the first row shows the scatter plot of the NSE efficiency; second row, the NSE cumulative distribution of the behavorial and non behavoiral classes indicating the Kolmogorov-Smirnov statistics D. Note: for each parameter, the point that is shown in the parameter space is the average of the mask (raster values) weighted by the class percentage for that specific run.

### 4.1.4   Sensitivity analysis summary

Table 6 and 7 gives the parameter sensitivity ranking of the three models according to the Kolmogorov-Smirnov test statistics
D, the results of the three models resulted in somehow similar conclusions. In the case of the Gardon, parameters of the model





1. Scatter plots

2.Cumulative distribution of the behavorial and non behavorial classes

**Figure 5.** RSA of the four GR4H parameters on the two study catchements (left column: Gardon, right column: Atrdeche). For each catchment, the first row shows the scatter plot of the NSE efficiency; second row, the NSE cumulative distribution of the behavorial and non behavoiral classes indicating the Kolmogorov-Smirnov statistics D

that affects the transfer are sensitive ($c_{tr}$ for SMASH; $x_3$ for GR4H and $C_{kss}$ for MARINE). Ardeche on the other hand has parameters that affects the production components of the model as generally sensitive ( $c_p$ for SMASH; $x_1$ for GR4H and $C_k$





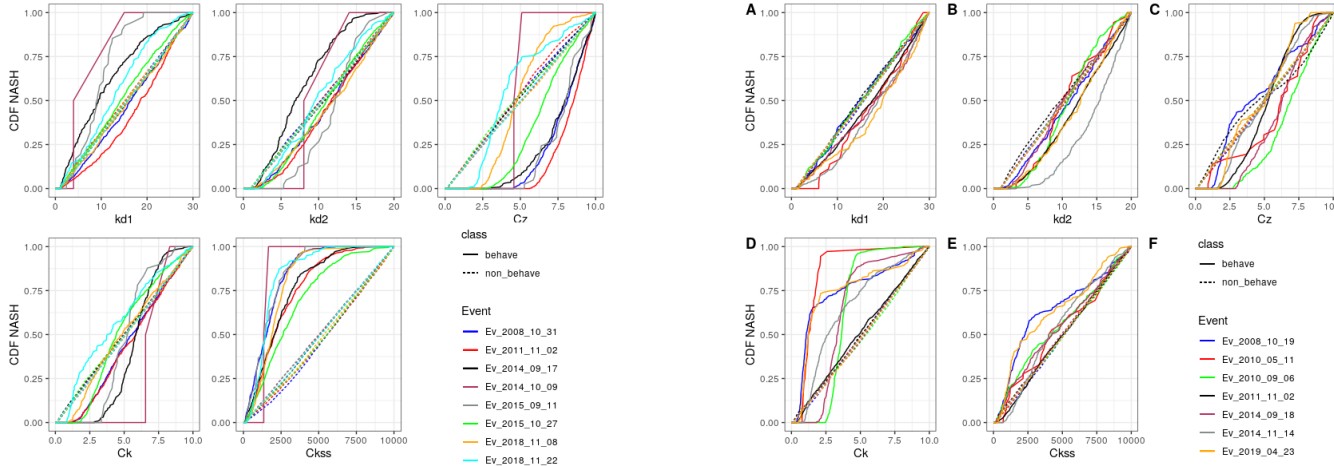

**Figure 6.** MARINE sensitivity analysis result showing the cumulative distributions of the behavorial and non behavorial classes of the five parameters for Gardon (left) and Ardeche (right)

for MARINE). As Beven (2001) rightly pointed out, the insensitivity of a particular parameter within a certain range might be due to inactivation of the model component associated with that parameter.

### 4.2 Response surface and functionning points

Figure 7, 8 and 9 shows the response surface plot of the three study models' parameters; SMASH, GR4H and MARINE respectively, for the two study catchments. The first row of Figure 7 shows the response surface resulting from spatially uniform parameters of SMASH model, while the second row is for the SMASH model, but for masked parameters. As the number of free parameters for both models is four (five) for SMASH and GR4H (MARINE), the visualization of the plot is complex at this high dimension, and so the plots are shown in 2D of two parameters per plot. The response surface highlights the complexity the calibration algorithm is subjected to in the search of the minima of the cost function during the optimization process. For all the models, the response surface indicates the existence of multiple peaks in multiple locations within the parameter space. This highlights the need for efficient algorithms able to detect global optimum during the calibration of the model parameters. The diamond shaped points shown on the plots indicate the functioning points obtained with the respective calibration algorithms. The fact that, these points lie within the hills/peaks, shows that the algorithms are able to locate 'sufficient minima', although few parameters are stuck at the bound. An exception is in the case of the masked calibration, although some of the values (eg $k_s$) lie outside the hills, it is possible the actual values has been distorted by the average, as a reminder, what we show on the plots are the spatial averages of the parameters. Again, the equifinality issue encountered in these optimizations as highlighted by the response plots.





**Figure 7.** Response surface of the parameters of SMASH on the two study catchments (left column: Gardon, right column: Ardeche). For each catchment, the first row shows the response plot of SMASH uniform parameters in pairs; while the second row, the response plot of SMASH masked parameters in pairs. The black diamond point shows the functioning point obtained by calibration.

## 4.3 Calibration and validation

### 4.3.1 SMASH

The result of the calibration of the SMASH model parameters is given in Table 8 for the two study catchments.





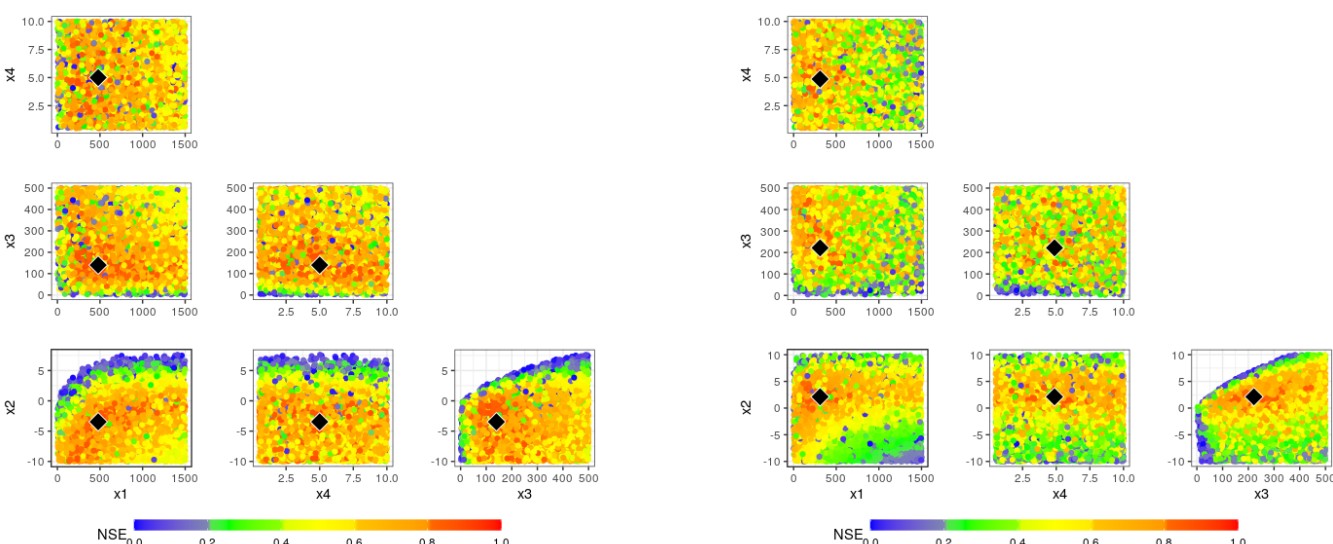

**Figure 8.** Response surface of the parameters of GR4H in pairs, on the two study catchements (left: Gardon, right: Ardeche. ). The black diamond point shows the functioning point obtained by calibration.

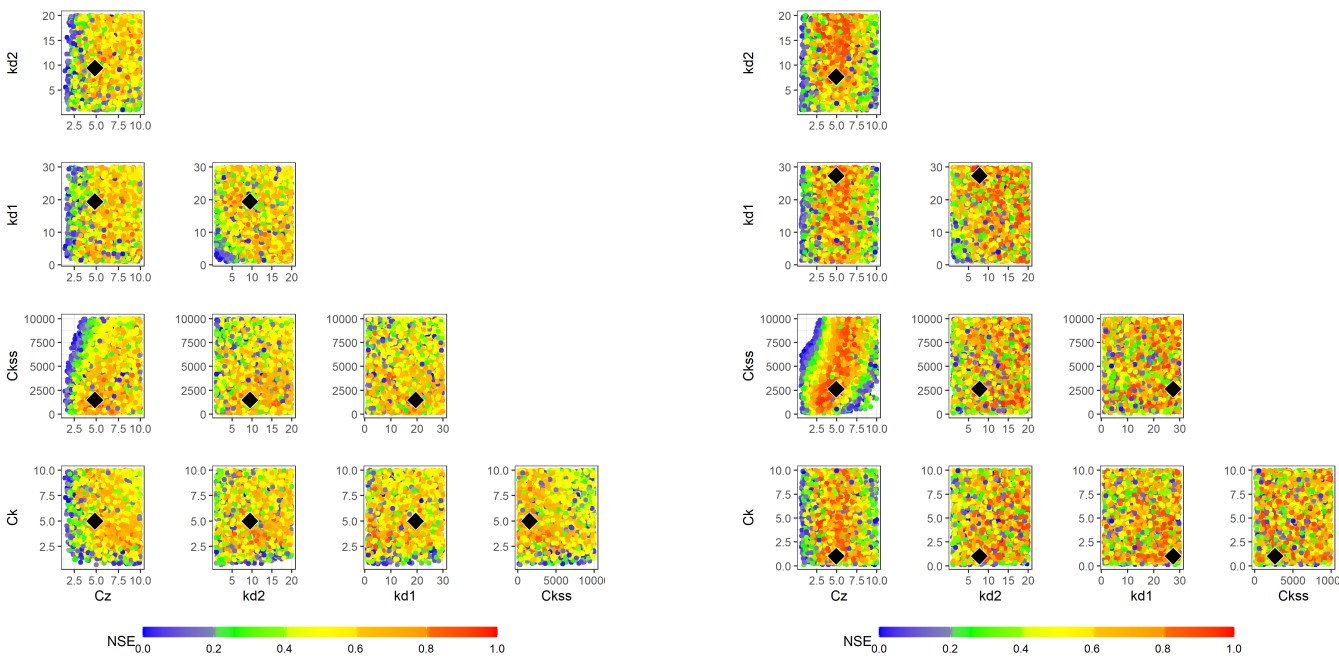

**Figure 9.** Response surface of the parameters of MARINE model for two selected events. Left (Gardon Ev 2015-10-27), Right (Ardeche Ev 2011-11-02) The black diamond point shows the functioning point obtained by calibration.





**Table 8.** SMASH parameter sets, calibration and validation NSE obtained using split-sample test for the catchments using split test. For each parameter, the mean and standard deviations of its map are shown

| Catchment | Period | $c_p$ | $c_{tr}$ | $v$ | $k_s$ | NSE calibration | NSE Validation |
|-----------|--------|-------|----------|-----|-------|-----------------|----------------|
| Ardeche | P1 | 164.5±127 | 359.0±88 | 4.64± 0.03 | 3.93± 0.5 | 0.873 | 0.839 |
|  | P2 | 203.0±85 | 365.4± 143 | 4.65± 0.03 | 1.33± 0.3 | 0.914 | 0.881 |
| Gardon | P1 | 1514.3±112 | 332.0±119 | 4.95±0.02 | 1.11±1.5 | 0.860 | 0.788 |
|  | P2 | 1193.6±247 | 262.9±121 | 4.89±0.03 | 1.05±1.1 | 0.776 | 0.737 |

The class by class (mask) calibration efficiencies for the two periods vary for the two catchments but both are more than 0.7. The resulting temporal validation efficiencies are also relatively high. Ardeche presents better calibration/validation efficiencies than the Gardon catchments. The maps resulting from the calibration are given in Figure 10 for both periods (P1 and P2) and their summaries in Table 8. The results for the Gardon (left) shows that the calibrated reservoirs capacities $c_p$ and $c_{tr}$ changes in magnitude with the calibration period (both are smaller in period 2), whereas the routing parameter $v$ remains fairly stable (as found in Jay-Allemand et al. (2020)). The converse is true in the case of Ardeche for $c_p$ and $c_{tr}$. The $k_s$ parameter however decreased in period 2 for both catchments. Jay-Allemand et al. (2020) has observed the same difference while studying the Gardon catchment under fully distributed calibration and has concluded that the differences is a result of different rainfall pattern between the two periods rather than from the calibration algorithm.

#### 4.3.2 GR4H

In the case of the calibration of the GR4H model on the two catchments, the parameters and efficiencies obtained both in calibration and validation are shown in Table 9. All calibration and validation efficiencies are higher than 0.7. In the case of Ardeche, there is relative stability/robustness in the calibration and validation efficiencies. The groundwater exchange coefficient $x_2$ are positive in both calibration periods for the Ardeche (export), while they are negative (import) in the case of Gardon. According to this model, positive values show water import, while positive values indicate water export.

**Table 9.** GR4H parameter sets, calibration and validation NSE obtained using split test for the catchments using split test.

| Catchment | Period | $x_1$ | $x_2$ | $x_3$ | $x_4$ | NSE Calibration | NSE Validation |
|-----------|--------|-------|-------|-------|-------|-----------------|----------------|
| Ardeche | P1 | 310.6 | 2.12 | 221.6 | 4.87 | 0.868 | 0.849 |
|  | P2 | 216.2 | 1.37 | 311.6 | 3.89 | 0.899 | 0.868 |
| Gardon | P1 | 478.5 | -3.46 | 139.9 | 5.0 | 0.908 | 0.835 |
|  | P2 | 230.4 | -6.49 | 136.1 | 4.33 | 0.777 | 0.733 |

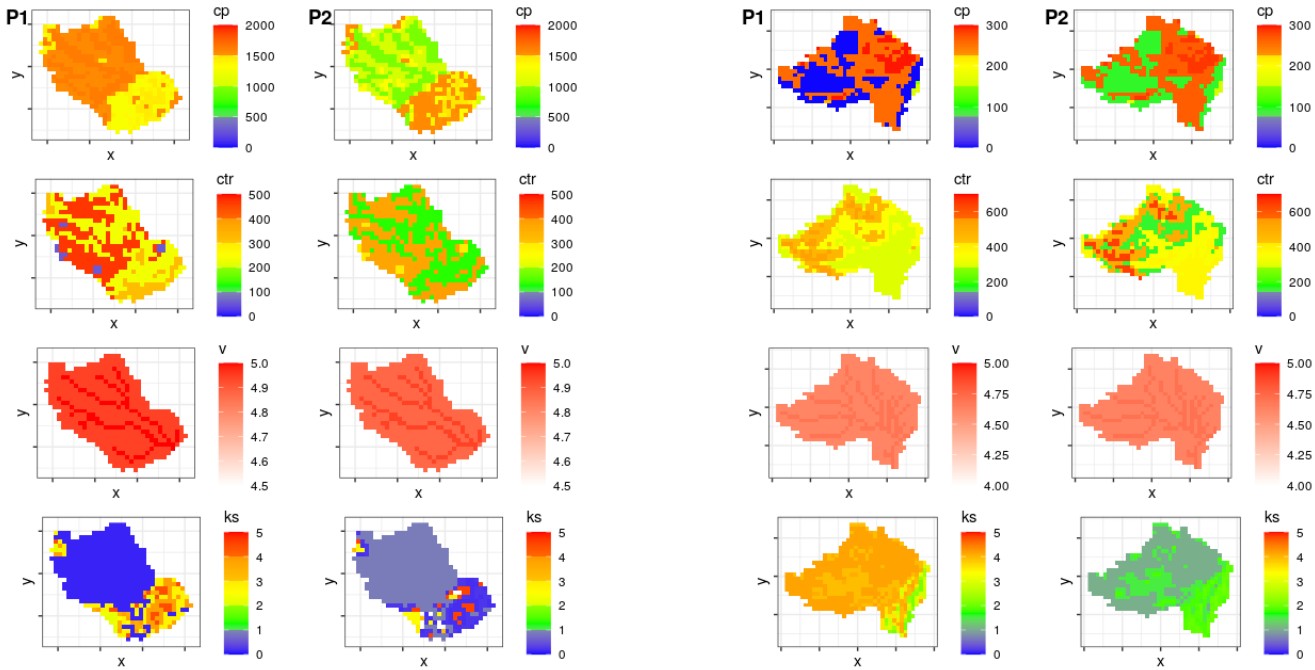

**Figure 10.** Maps of the SMASH calibrated parameters for Gardon (left) and Ardeche (right)

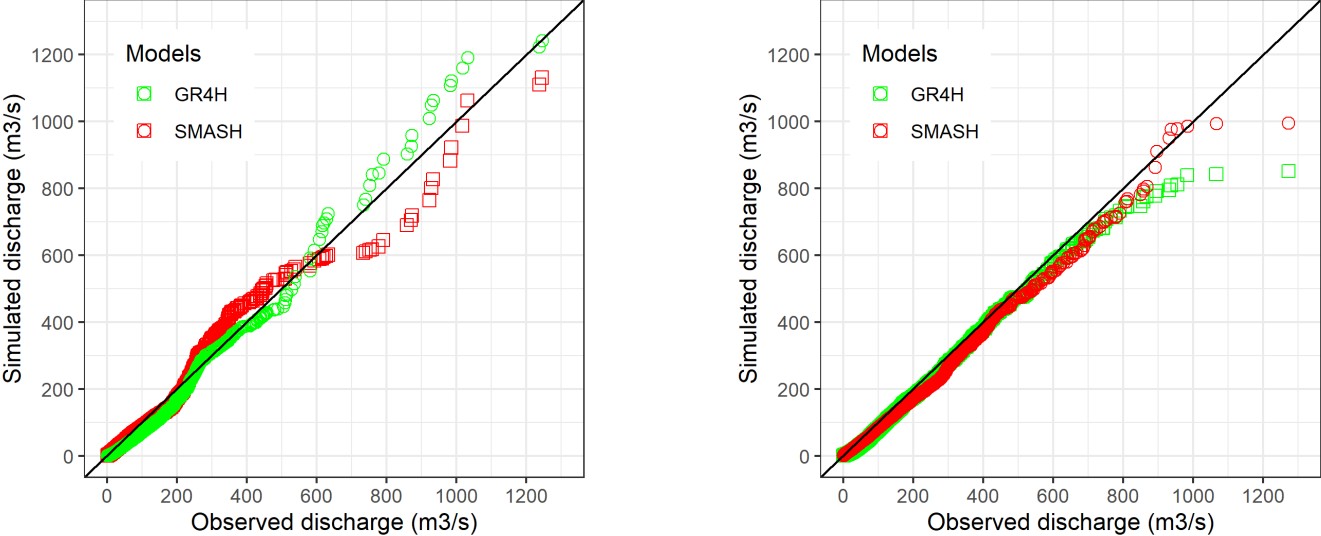

**Figure 11.** QQ plot of observed and simulated discharges in validation for period 1 (2006 to 2013) simulated with the calibrated parameters obtained from period 2 (2012 to 2019). Simulations obtained with the continuous SMASH and GR4H models for (left) Gardon and (right) Ardeche.



### 4.3.3 MARINE

The calibration of this event-based model followed the procedure described in the methodology, the events were divided into
period 1 and period 2 events and each group was calibrated together by minimizing the cost function of $1 - NSE$ using the
events. The resulting global efficiencies are presented in Table 10 for both catchments. Event specific NSE (not shown here)
has an average of 0.87 and 0.78 for the Gardon events of period 1 and period 2 respectively. Subsequently cross validation was
done using the calibrated parameters from each group. This is to ensure the same calibration validation approach is done for
the three models. During the calibration of the model and for each set, at least 10 starting points in the parameter sets were
tested in order to ensure that the optimization is not stuck to local optima within the response surface.

The period 1 and period 2 events of the Gardon catchment resulted in very similar values, except the $C_z$ parameter which is
almost twice in period 1 compared to period 2. For the Ardeche, higher calibration efficiencies are obtained compared to the
Gardon, although the parameters between the two periods are dissimilar.

Validation efficiencies in terms of the Nash are presented in Table 11 for both catchments. The efficiencies are event depen-
dent. For Gardon, NSE as high as 0.91 is obtained and as low as 0.09 , with the average of 0.58 for the eight (8) events. The two
Nov 2018 events presenting the least efficiencies have the least observed peak magnitudes (655 and 809) compared to the max
of 1356 $m^3/s$ observed with the Oct 2015 event. It is thus possible that the soil thickness coefficient used (8.0) is too large for
these events. In the case of Ardeche, the NSE in validation is also event dependent, the min/max obtained is 0.47/0.87 with an
average of 0.77. Finally, the temporal performance decrease in validation is smaller in Ardeche (from 0.96 to 0.77 on average)
compared to Gardon (0.85 to 0.58)

**Table 10.** Catchment parameter sets and NSE for multiple event calibration based on split test using MARINE

|         | Period | $K_{D1}$ | $K_{D2}$ | $C_Z$ | $C_k$ | $C_{KSS}$ | Global Nash | No of Events |
|---------|--------|----------|----------|-------|-------|-----------|-------------|--------------|
| Gardon  | P1     | 19.42    | 9.45     | 8.0   | 4.99  | 1497      | 0.88        | 2            |
|         | P2     | 19.44    | 9.43     | 4.83  | 4.99  | 1500      | 0.82        | 6            |
| Ardeche | P1     | 27.39    | 7.73     | 4.91  | 1.02  | 2638      | 0.97        | 4            |
|         | P2     | 18.43    | 14.57    | 2.23  | 4.36  | 1719      | 0.95        | 3            |

### 4.3.4 Comparison in calibration and validation

Considering the two continous models, SMASH and GR4H, the global efficiencies in time obtained in calibration are similar.
However, they are slightly lower than SMASH for the Ardeche catchment, while they are higher for the Gardon catchment.
The temporal validation efficiencies are however catchment dependent, they are higher in period 1 for GR4H compared to
SMASH, while being lower in period 2. But summing up on both periods and on both catchments, MARINE has its efficiency
in validation decreased by around 25%, while SMASH and GR4H have a decrease of 5.2% and 4.8% respectively.





**Table 11.** NSE event performance criteria in validation of the outlet discharge for the study catchments. For each catchment, the events marked with (*) are period 1 events, while the others are period 2 events.

| Gardon | | | | Ardeche | | | |
|---|---|---|---|---|---|---|---|
| Event | MARINE | SMASH | GR4H | Event | MARINE | SMASH | GR4H |
| Ev_10_11_2008* | 0.82 | 0.90 | 0.66 | Ev_2008_10_19* | 0.79 | 0.92 | 0.74 |
| Ev_01_11_2011* | 0.66 | 0.91 | 0.61 | Ev_2010_05_11* | 0.47 | 0.68 | 0.16 |
| Ev_16_09_2014 | 0.50 | 0.66 | 0.07 | Ev_2010_09_06* | 0.73 | 0.66 | 0.28 |
| Ev_09_10_2014 | 0.69 | 0.72 | 0.68 | Ev_2011_11_02* | 0.84 | 0.94 | 0.72 |
| Ev_10_09_2015 | 0.91 | 0.60 | 0.16 | Ev_2014_09_18 | 0.86 | 0.38 | 0.71 |
| Ev_27_10_2015 | 0.79 | 0.58 | 0.67 | Ev_2014_11_14 | 0.87 | 0.80 | 0.55 |
| Ev_22_11_2018 | 0.09 | 0.81 | 0.82 | Ev_2019_04_23 | 0.85 | 0.93 | 0.89 |
| Ev_08_11_2018 | 0.19 | 0.91 | 0.93 | | | | |
| **Average** | 0.58 | 0.76 | 0.58 | **Average** | 0.77 | 0.76 | 0.58 |

With regards to the obtained parameters, the same decrease of the production reservoir capacity $x_1$ period two for the Gardon is obtained as observed with SMASH, Ardeche however presents a decrease of $x_1$ in the second period compared to the observed increase with SMASH. Both models however resulted in larger $x_1$ for the Gardon compared to Ardeche (although the difference is much larger with SMASH). The routing reservoir $x_3$ is however larger in the case of Ardeche. The ground water exchange are positive for Ardeche, indicating water import, while negative for Gardon, indicating water export. Finally the time base of the unit hydrograph $x_4$ is between 4 and 5 hours for both catchments.

Compared to the values of the saturated hydraulic conductivity $ks$ parameter obtained with MARINE (see section 4.3.3), the values obtained here with SMASH seamed rather low, although both use a Green and Ampt infiltration model, but MARINE is calibrated specifically on flood events and so the activation of Hortonian flow would be different compared to SMASH that is calibrated on both high and low flow events. Also, in addition to the surface flow, a component to allow for lateral subsurface flow is available in MARINE, hence the contribution to the flood hydrograph is a combination of the two. The choices made in SMASH with regards to the value of the porosity in the Green and Ampt model (kept at a value of 1), and suction values obtained from the map of the soil texture according to Rawls and Brakensiek might also affect the final calibrated value of the $ks$. Note that the values with MARINE are for the multiplicative constant $C_k$ which in principle is multiplied to the gridded values of the saturated conductivity maps.

## 4.4 Comparison at event scale

In this section, the event scale performance of the models is compared. This is done through the signatures of the simulated discharge and the simulated soil moisture. While the simulated hydrographs are compared with the observed hydrographs through the computed metrics, the soil moisture is compared to the outputs of the SIM2 model.





### 4.4.1 Discharge Simulation

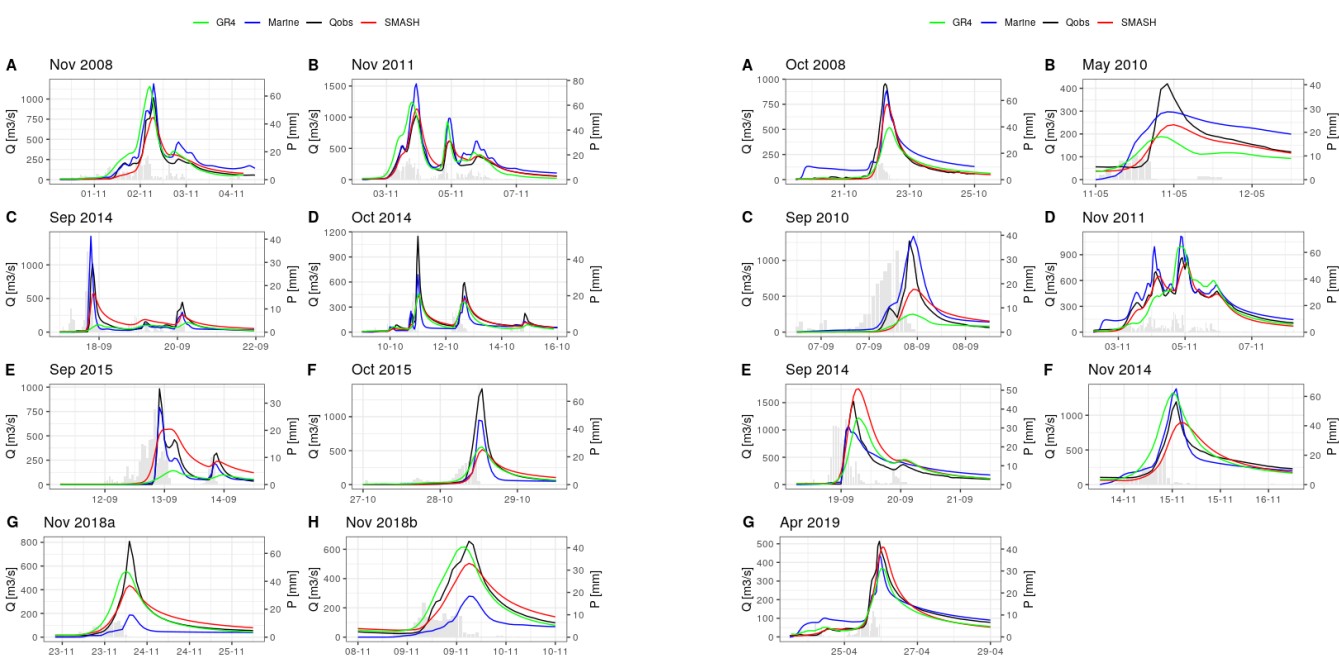

**Figure 12.** Flood events, measured at the outlets, simulated with MARINE, SMASH, and GR4H for Gardon (left) and Ardeche (right)

Figure 12 compares the simulated discharges with the three models against the observed discharges for Gardon (left) and Ardeche (right). The performance of all the models seems to be fair, and the superiority of the models depends on the event. It is however difficult to judge objectively from the figures of the hydrographs. In order to judge objectively, different metrics have been computed and are shown for both catchments in Figure 13 and 14. The performance of the models is therefore judged and discussed according to these metrics in the following paragraphs.

First, looking at Figure 13 in terms of NSE, the superiority in performance of the models over one another is quite event dependent, however, for most of the events in the Gardon catchment, SMASH has better NSE values. The average NSE for the eight events is 0.76 for Gardon against 0.58 for both MARINE and GR4H. For Ardeche catchment, MARINE is slightly better with 0.77 average against SMASH with 0.76. GR4H remained the lowest with 0.58 average. In terms of the NSE, SMASH performed better compared than the other models, while GR4H is the least.

An alternative to NSE is the KGE metric. Although NSE is used in calibration, the KGE criteria is also used to evaluate the performance. This metric gives an aggregated measure in performance in terms of the correlation, mean (water balance), and flow variability bias. Considering Gardon, SMASH has an average of 0.65 against 0.48 for GR4H and 0.44 for MARINE. For Ardeche on the other hand, SMASH remains better for most of the events, compared to the other models. The average





for SMASH is at 0.73 compared to 0.67 and 0.53 for MARINE and GR4H respectively. Again for the Ardeche, MARINE outperformed GR4H on average.

The three components of the KGE, also reveals some relevant information on the performance of the models. In terms of the correlation coefficient $r$, which assesses the error in terms of the shape and timing of the hydrographs, all the models have

relatively high values. MARINE however has on average better performance based on this criteria in both catchments (0.94 and 0.96). GR4H remains the least in both (0.83 and 0.89). With this high average, it can be inferred that all the models are capable in terms of reproducing the shape and timing of the hydrographs. $\beta$ measures the bias in terms of the mean (water balance). SMASH has the least bias compared to both catchments (1.08 and 0.99), while MARINE has the highest bias (0.78 and 1.13). Finally the measure of bias in the flow variablity $\alpha$, indicates that for most of the events, SMASH has the least bias.

On average however, the bias is the same for GR4H an MARINE.

Other indicators to objectively compare the models are shown in Figure 14 and given in Table C2 and Table C4 for Gardon and Ardeche respectively. In terms of the percentage difference in peak magnitude, PPD, MARINE model approximates the observed peak better than the other models for most of the events in the two catchments. The difference in the timing of the observed and simulated peak is also less observed with MARINE simulations, SMASH on average has less differences

compared to GR4H. The percentage difference between the observed and simulated peak at the time of the observed peak measured by the SSPD criteria indicates more accurate simulations with MARINE. SMASH is yet, more accurate than GR4H based on this criteria. This criteria is relevant because, it is important to know not only the difference between the observed and simulated peak, but also what peak is simulated at the time the observed peak occurs. Lastly, the runoff coefficient (CR), measures the ratio of the total flow over the total precipitation. SMASH gives the closest CR to the observations for most of

the events in the two catchments compared to the other models, it is also the closest to the observations in terms of the average of the CR for both catchments. GR4H closely follows, while MARINE is the least of the two models for both catchments.

Inferring from the results, the event based MARINE has better performance with regards to the peak simulation and timing, followed by SMASH. However, in terms of the volume of water exported and water balance, SMASH performed relatively better and is followed by GR4H.

Although both SMASH and GR4H models used the same conceptual production reservoir thickness, the production reservoir in SMASH (used in this study) is filled according to the Green and Ampt infiltration function; (infiltration rate equals the rainfall intensity provided ponding doesn't occurs, when it does, the infiltration excess is transferred). GR4H on the other hand is based on the saturation mechanism in which rainfall excess occurs only after saturation. This, in addition to the distributed nature of SMASH, could partly explain why SMASH outperformed GR4H in terms of the indices of peak magnitude and timing.

This is despite the fact that GR4H, by construction, has more complexity in terms of processes represented and formulations used, including a non conservative exchange term (parameter $x_2$) (see A1). MARINE, apart from the physical basis, processes represented, and complexities in the formulations, is simply calibrated over flood events only. The continuous models are however, calibrated on all the flows (both low and high) and would therefore perform better in terms of the volume of the flood.




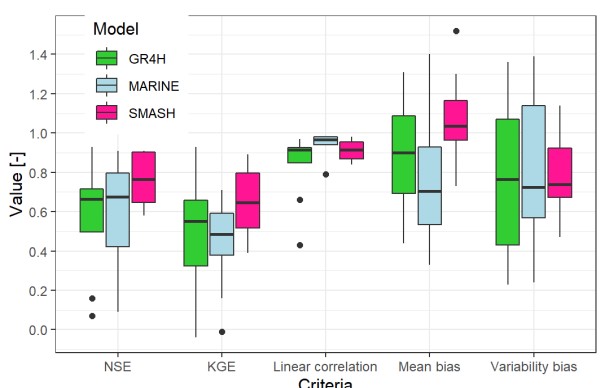
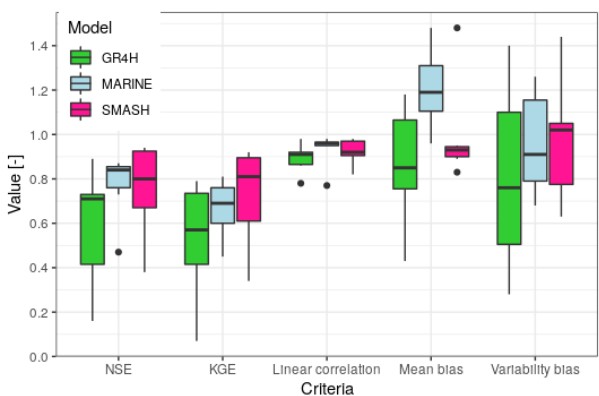

**Figure 13.** Integrated metrics of the simulated hydrographs in validation, of the three models for Gardon (left) and Ardeche (right). Metrics are computed for the events shown in Table 5

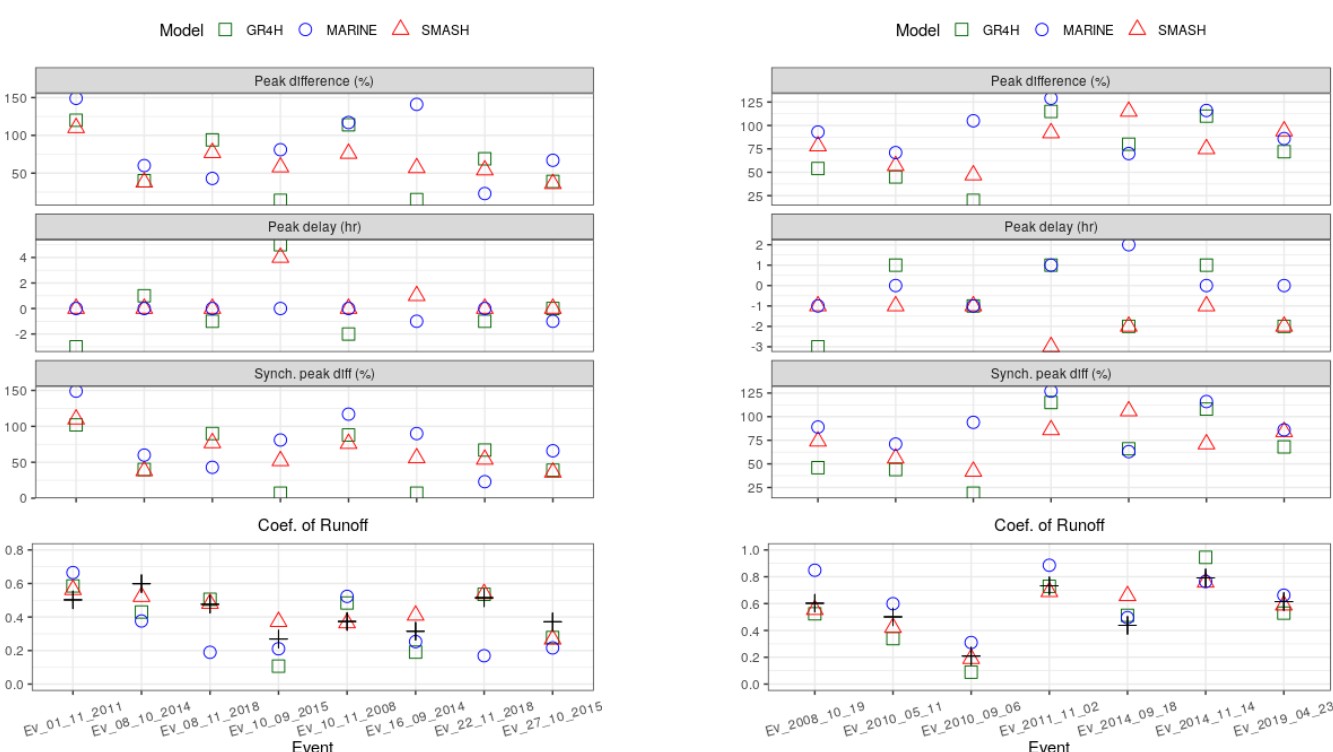

**Figure 14.** Comparison in validation of the SMASH, GR4H and MARINE in terms of some hydrological signatures; the Percentage Peak difference (PPD), the time difference of the peak (PD), the Synchronous Percentage of the Peak Discharge (SSPD) and the runoff coefficient (CR), Gardon (left) and Ardeche (right) Black cross: observed runoff coefficient.





### 4.4.2 "Soil moisture" comparison

The spatially averaged time series of the soil moisture predicted by the three models is shown in Figure 15. In case of the two distributed models, SMASH and MARINE, the spatial average over the area of the catchment at the hourly temporal scale is shown. The spatial average of the soil moisture outputs of the two SIM products, SIM1 and SIM2 are also shown. In the case of SIM1, which is used for initialization of the MARINE model, the single value per event (spatial average) corresponding to the beginning of the event is shown, while for SIM2, which is used for comparison, the daily series (available for this study) is

shown at 06:00hr of every day for the event duration.

First, the soil moisture output of SIM1 (shown at the beginning of every event) is always lower in amplitude compared to the output of SIM2. While the former discretizes the soil into three layers, the middle layer corresponding to the root zone, the later discretizes into 14 layers, the layers between 10-20 cm corresponding to the root zone.

Using the SIM2 series as benchmark for comparing the three models, MARINE performed best in terms of both the dynamics

and amplitude of the soil moisture in both catchments. It is closely followed by GR4H model, while SMASH is the least in comparison to the other models. To assess the goodness of fit between soil moisture series of the three models in comparison to that of the SIM2 (shown in Figure 15), Figure 16 summarizes the root mean square error (RMSE) on the eight(seven) events of Gardon(Ardeche), shown on the left and right of the figure respectively. For both catchments, MARINE is the most accurate (lowest RMSE), followed by GR4H (looking at the median). In the case of Ardeche (right), the 0.75 quantile is lower than the

0.25 quantile of the other two models.

Looking at the SMASH model, we see that in the case of Gardon catchment, the series remained relatively flat, and the response between rainfall events is very week. Better response are however observed in the case of Ardeche compared to Gardon. This could be possibly explained by the size of the calibrated production reservoir capacity $c_p$ of the two catchments. Relatively large capacities of $c_p$ (1500 and 1200 mm for period 1 and 2 respectively ) for Gardon against (164 and 200mm)

for Ardeche are obtained. The depletion of the smaller capacity production reservoirs after or between rainfall events would be faster compared to the larger ones. Interestingly, GR4H calibration resulted in much smaller $c_p$ for Gardon (480 and 230mm for period 1 and 2 respectively) compared to SMASH.

The difference in performance in the soil moisture outputs could be explained by the complexities and processes represented in each of the models. MARINE, in addition to the surface flows (overland and in the channels), subsurface lateral transfers

are represented using an approximation of the Darcy's law. Therefore, although evaporation is deemed negligible at the event scale, thus not represented, the lateral flows contributes to the emptying of the soil reservoir and hence the faster and sharper decline between and after rainfall events. In addition to this, being a physical model, soil surveys are used as the basis for the soil depths (corrected by a multiplicative factor $c_z$). This makes the process and soil moisture variation potentially closer to the real physical phenomena, unlike in the other two models in which the depths are fully conceptual - and more or less free to

vary in space.

Although both SMASH and GR4H are emptied by the same evaporation function (see Equation A2), GR4H soil reservoir is also emptied by a percolation leakage. This percolation leakage although weak, given the power law involved, is an added





complexity in the model that might have resulted in the faster response between rainfall events compared to SMASH. The process of soil emptying of the SMASH (distributed) model is thus more likely to be weaker than that of GR4H (lumped).

In the case of Gardon, the soil saturation of SMASH is generally lower than GR4H for most of the events. This is likely due to the size of the respective production reservoirs (1500mm for SMASH and 500mm for GR4H). Apparently, for the same rainfall signal, the soil moisture will be higher in the smaller sized reservoir. An emphasis of this can be seen in Ardeche catchment where SMASH soil moisture are higher for all the events. Interestingly, the production reservoir depth for this catchment is 160mm for SMASH and 300mm for GR4. Hence, SMASH saturation are higher (due to smaller capacity). The

optimized reservoir depth from the model calibration therefore affects the accuracy of the soil moisture estimation.

The controlability of the models also is different, although all the three models use the outlet discharge as the variable of interest in the calibration, MARINE model has constraints on its parameters using field data (soil survey and vegetation and land use) both in terms of their spatial distributions and their magnitude, although the later is corrected using some coefficients during calibration. The production reservoir is constrained by the soil thickness map, the Green and Ampt parameter (porosity,

hydraulic conductivity and suction) are all constrained using the soil classes derived from the soil texture. The subsurface transfer have also been constrained by the soil classes and finally the Manning friction in the kinematic wave routing formulation for overland flow by the land cover. This gives MARINE more constraints in its parameters thereby having parameters with some level of physical meanings rather than being simply artifacts of the calibration algorithm. The fact that SMASH uses the same maps during calibration doesn't offer as much constrains as in MARINE model. In fact, the use of the maps is only to

reduce the high dimensionality resulting from fully distributed calibration. The constraints are thus applied only on the spatial pattern rather than on their magnitudes as done with MARINE. Again, even the choice of the field data eg (soil surveys of thickness and texture) to use for the constraint on the spatial pattern of SMASH parameters is not as clear as that of MARINE, since the parameters of the later have some physical meanings compared to more conceptual nature of the SMASH parameters. The least constrain applied in terms of spatial pattern is thus on the GR4H model which is lumped, and thus rely solely on

the outlet discharge in the optimization process. Lastly, MARINE is also constrained using information from the SIM1 soil moisture output for its initialization.

To investigate the temporal evolution of the soil saturation, Figure 17 presents maps for two chosen events; Sep 2015 and Sep 2014 for Gardon and Ardeche respectively. The figure shows the maps of the cumulative rainfall in mm, the map of the soil moisture in % SIM2 (the reference) and those for the three competing models (SMASH, MARINE and GR4H). For each

model, two maps are shown, before and after the rainfall event. The maps reinforces the results seen in Figure 15, SMASH overestimates the soil moisture before and after the floods. Suprisingly, in the case of the Gardon catchment, at the end of the Sep 2015 event, different pattern of the soil saturation is observed. While the saturation is higher upstream of the catchment according to MARINE (mostly along the drainage networks), it is higher downstream according to SMASH.





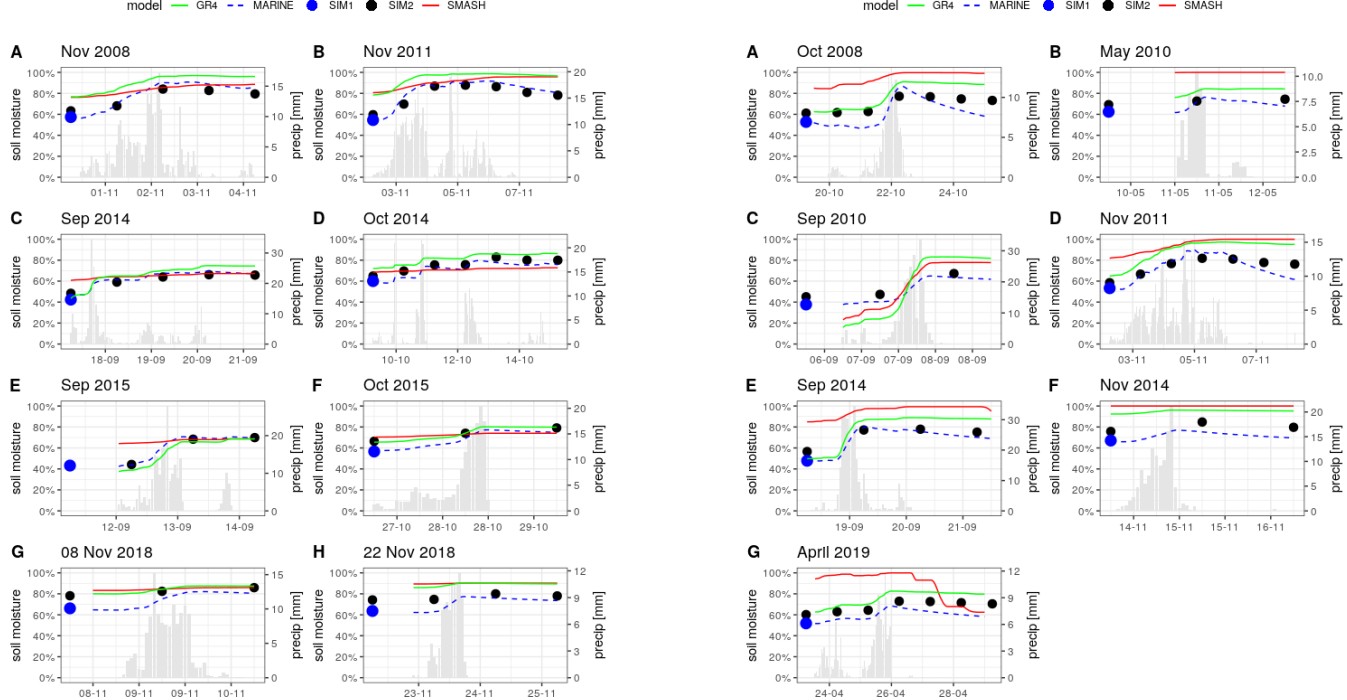

**Figure 15.** "Soil moisture (Internal signature)" time series, on average per catchment and event, simulated with MARINE, SMASH, GR4H, and the daily outputs of the SIM1 and SIM2 models for Gardon (left) and Ardeche (right)

### 4.4.3 Available storage

In this part, the sensitivity of the available storage to the perturbations of the model parameters is investigated for SMASH model. The available storage gives an insight into what volume is available to dampen or filter the rainfall signal given the behavior of the catchment, the parameters and the specific event. We try to investigate how the model parameterization affects the change of this important component. For each model run, corresponding to a vector of parameters sampled within the given range according to the classical RSA, the evolution of the soil saturation map at the beginning of an event and at the time of the maximum discharge for that event is considered, for simplicity, two considerations are made. First, the time of the observed peak is assumed to correspond to the time of maximum soil saturation. This becomes necessary as it is difficult to anticipate before hand, or to track the exact time of maximum saturation, given that 10,000 runs are made for a total period of 13 years with different parameters for each run, and the event duration is a very small fraction of that time. Secondly, the evolution of the available storage is considered between two time steps, the beginning of the event and the time of the maximum observed peak. Knowing that the model is spatially distributed, spatial average of the moisture saturation is considered for the analysis. This is coherent since the model parameters for this experiment are taken as spatially uniform. The events considered for this study are the same events described in Table 5.





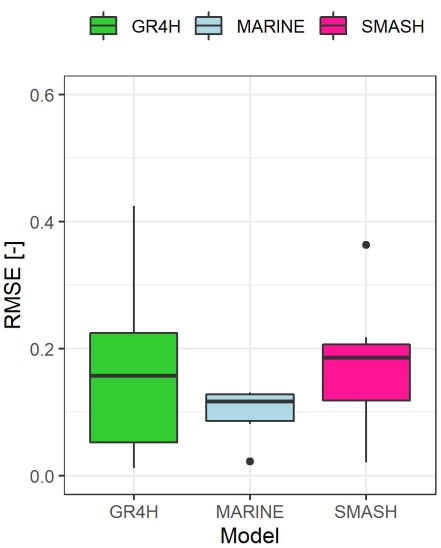
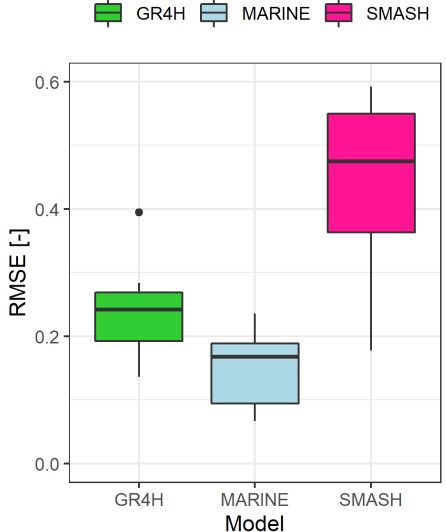

**Figure 16.** Boxplots of the root mean square error (RMSE) computed on the soil moisture series shown on Fig 15 for Gardon (left) and Ardeche (right). Optimum value of the RMSE is 0

The scatter plots resulting from the experiment are shown in Figure 18 for the seven events. For simplicity, the other model parameters are not shown and only the production reservoir capacity $c_p$ is shown. The scatter plots for the other parameters showed non-identifiability irrespective of the event considered and of the parameter value within the considered range. In the case of the $c_p$ parameter, apparent identifiability is observed for all the events. The points resulting from the multiple model runs converged to a line for each event, showing that irrespective of other values taken by the other model parameters, the change in available storage is the same for a particular value of $c_p$. For very small values of $c_p$, the change is very high (near 100%) and for high values, the change is comparatively low. This could be explained by the fact that, for the small values of $c_p$, the storage is inherently small, and so the input signal of rainfall will easily result in saturation, and hence a large change in the available storage. The resulting curves are interestingly different for the seven events, the May 2010 event shows the least change in available storage while the Nov 2011 shows the highest irrespective of the capacity $c_p$. These two events has the least and largest transported volume respectively.

# 5 Conclusions

The aim of the study was to understand how three models of varying complexities simulate the hydrological behavior of two flash flood Mediterranean catchments; Gardon at Anduze and Ardeche at Vogue, both located in the South of France. The methodology involved the investigation of global parameter sensitivity of the models, their efficiencies in calibration and validation, and the assessment of key hydrological signatures at the event scale. Finally the soil moisutre simulated by the





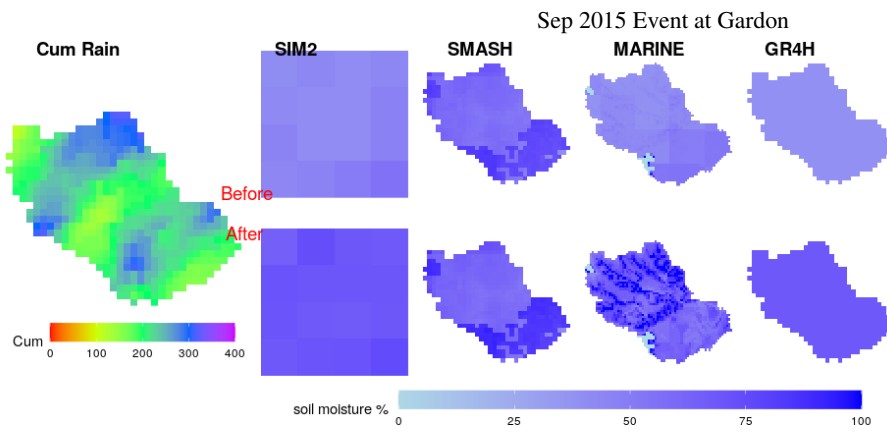

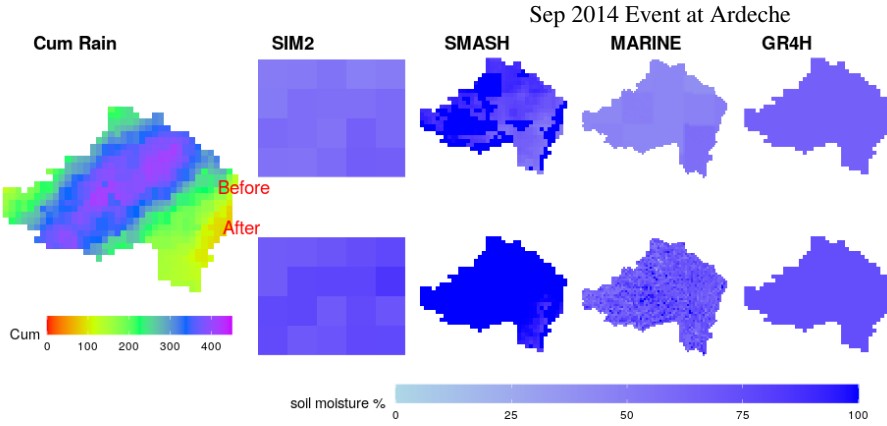

**Figure 17.** Cumulative rainfall in mm, and "soil moisture (Internal signature)" maps before and after some selected events, simulated with MARINE, SMASH, and GR4H. The daily outputs of SIM2 model is also shown. The events are Sep 2015 for Gardon (top) and Sep 2015 for Ardeche (down)

three models at the event scale were compared with the gridded soil moisture outputs of the hydrometeorological SIM model. The three hydrological models are the lumped conceptual model GR4H, spatially distributed conceptual model SMASH, and process oriented distributed model MARINE.

The invested methodology followed and the results obtained lead to the following conclusions. First, the global sensitivity analysis using RSA of the three models revealed contrasted parameter sensitivity to the same efficiency measure, and depending 635   on the catchment considered. In the case of SMASH model, the dimension reduction through the use of uniform parameters revealed that the $c_{tr}$ parameter that controls the transfer as the most sensitive for Gardon, and routing velocity $v$ for Ardeche, while under masked parameters, $c_{tr}$ remained the most sensitive for Gardon, and $c_p$ for Ardeche. Concerning the lumped GR4H however, for both catchments, the time base of the unit hydrograph $x_4$ is the least sensitive, while the ground water





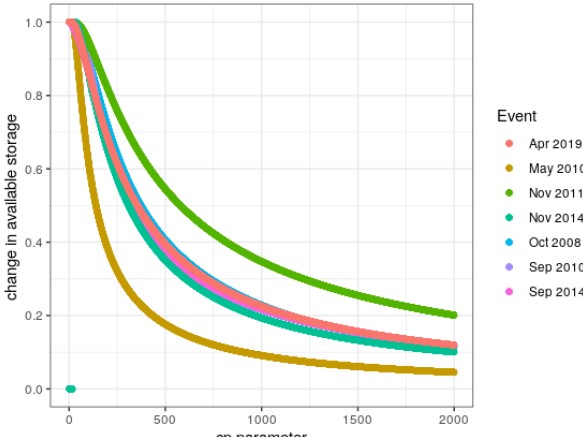

**Figure 18.** Scatter plot of the sensitivity of the change in available storage between the event beginning and the time of the maximum observed peak, to the production reservoir capacity $c_p$. Results are shown for the seven study events at Ardeche catchment within the 2006 to 2019 study period.

coefficient $x_2$ ("non conservative" flow component) is the most sensitive. For the event based MARINE, the coefficient applied
to the lateral subsurface flow, $C_{kss}$ emerged as the most sensitive for Gardon, while the correction coefficient of the hydraulic conductivity $C_k$ emerged as the most sensitive in the case of Ardeche.

The response surface enhanced the existence of equifinality and highlighted the difficulty of the calibration problem especially for a distributed model to reach sufficient minimum of the cost function. For all the models, the functioning points obtained with the calibration algorithms are within hill regions of the response surface.

All the three models showed good calibration and validation efficiencies. Their performances were however, generally better on Ardeche compared to Gardon. In calibration, MARINE achieved the highest efficiency and is followed by GR4H. Although all the three models showed decrease in efficiencies at temporal validation, GR4H was more robust. Regarding the parameter stability between the two periods, all the models showed some differences between the calibrated parameters of both periods.

At the event scale, seven events and eight events of contrasted behaviors on Ardeche and Gardon respectively were selected
to compare the performance of the three study models on the simulated discharge and the soil moisture pattern. Indices of discharge simulation showed that, the event based MARINE has better performance with regards to the peak simulation and timing, and is followed by SMASH. However, in terms of the volume of water exported and water balance, SMASH performed relatively better and is followed by GR4H.

Using the soil moisture output of the SIM2 model as benchmark for comparing the simulated moisture by the three models
at the event scale, MARINE emerged as the most accurate in-terms of both the dynamics and amplitude of the soil moisture in both catchments (recall MARINE soil water content is initialized with SIM1). It is closely followed by GR4H model, while SMASH is the least compared to the other models. The SIM2 product from SIM model revealed to be a valuable information to assess internal dynamics of model states.





The difference in the model performances could stem from differences in the levels of complexity of the models, the pro-
cesses described and the constrains of the models, and thus highlights the need for future improvements in the models and
calibration methods. These include improvements on the components of vertical and lateral flow of the MARINE model, as
well as those of the SMASH platform along with its calibration and assimilation algorithms. In addition to this, improved
constraints on the patterns and magnitudes of SMASH parameters, including those of the Green and Ampt model, are required
to fully utilize its capacity especially under intense rainfall events. An extended multi-catchment comparison would enable
a more fair assessment and model inter-comparison. For soil moisture comparison, using other products in addition to SIM,
similar to the work of Eeckman et al. (2020) could provide more robust conclusions. Finally this study paves the way for
extended model hypothesis testing and intercomparison in the light of multi-sourced signatures to asses/discriminate internal
model behaviours, given multiple plausible parameters sets potentially corresponding to contrasted functioning points, hence
model components activation/interplay at given model structure for instance.

**Appendix A: Model Formulations**

**A1  SMASH**

**A1.1  GR water balance operators**

Initially proposed for a minimal complexity description of catchment water balance functioning, based on empirical model-
ing, the "GR loss model" Edijatno and Michel (1989) considers a production reservoir $\mathcal{P}$ of maximum depth $c_p$ and water
level $h_p$ and is recalled here for clarity. The neutralized rainfall and evaporation are respectively denoted $P_n$ and $E_n$. If
$P \geq E$ then $P_n = (P - E)$ and $E_n = 0$ and $\mathrm{d}h_p = \left(1 - \left(\frac{h_p}{c_p}\right)^2\right) dP_n$. If $P < E$ then $E_n = E - P$ and $P_n = 0$ and $\mathrm{d}h_p = -\frac{h_p}{c_p}\left(2 - \frac{h_p}{c_p}\right) dE_n$ . Assuming a stepwise approximation of the inputs $P(t)$ and $E(t)$ the temporal integration of these or-
dinary differential equations, enabling analytical solutions (calculation given in Edijatno (1991)), as reported by Perrin et al.
(2003) gives the infiltrating rainfall $P_p$ and the actual evapotranspiration from the reservoir store $E_p$:

$$P_p = c_p \left(1 - \left(\frac{h_p}{c_p}\right)^2\right) \frac{tanh\left(\frac{P_n}{c_p}\right)}{1 + \left(\frac{h_p}{c_p}\right) tanh\left(\frac{P_n}{c_p}\right)} \tag{A1}$$

$$E_p = h_p \left(2 - \frac{h_p}{c_p}\right) \frac{tanh\left(\frac{E_n}{c_p}\right)}{1 + \left(1 - \frac{h_p}{c_p}\right) tanh\left(\frac{E_n}{c_p}\right)} \tag{A2}$$

As remarked in Jay-Allemand et al. (2020), $h_p$ is the water level of the production reservoir at the begining of a time step
$\Delta t$ and $P_p$ and $E_p$ are the amount of water gained or lost over $\Delta t$ and used to update $h_p$ before the next time step.
This is the water balance scheme of GR4 where the state $h_p$ and parameter $c_p$ are respectively denoted $S$ and $x_1$.





## A1.2 Green and Ampt infiltration

Applying Darcy's law, Green and Ampt (1911) proposed a simplified physical model for water infiltration from a ponded surface into a deep homogeneous soil with uniform water content. The Green and Ampt model approximates the curved soil moisture profiles of the wetting front that result in practice, and from solution to Richard's equations, as a sharp interface with

saturation conditions $\theta = \theta_s$ above the wetting front, and initial moisture content $\theta = \theta_i$ below the wetting front. The initial moisture content is assumed to be uniform over the entire depth. The infiltration $i(t)$ writes as:

$$
i(t) = \begin{cases} r(t) & t \le t_p \\ K_s(1 + \psi \frac{\triangle \theta}{I(t)}) & t > t_p \end{cases} \tag{A3}
$$

Where $r(t)$ is the rainfall rate (m/s), $t_p$ is the time to ponding (s), $K_s$ is the saturated hydraulic conductivity (m/s), $\triangle \theta$ is the change in the volumetric water content (m/m), $\psi$ is the soil suction and $I(t)$ is the cumulative infiltration depth (m).

This model is used in MARINE event-based model (Roux et al., 2011).

It is also implemented in SMASH, following the algorithm presented in (Chow et al., 1998) involving a classical Newton-Raphson algortihm to solve $\triangle \theta$ from non linear Green and Ampt integrated in time (Mein and Larson, 1973), and with parameters explained in 3. Hence, the production reservoir $\mathcal{P}$ of maximum capacity $c_p$ (and porosity is simply set to $\eta = 1$) is filled by the infiltrating rainfall obtained form Equation A3, and is emptied by the actual evaporation $E_p$ obtained from

Equation A2.

## A1.3 Transfer

The Transfer function is represented by a reservoir of capacity $c_{tr}$ and actual level $h_{tr}$, and models the fast flow, it is supplied by the excess flow after production step (GR evaporation A2, infiltration A3). The time evolution of the actual reservoirs levels thanks to the mass conservation gives the flow rate $q_r$ from the fast reservoir at each time step such that:

$q_r(t) = h_{tr}(t) - (h_{tr0}^{-4} + c_{tr}^{-4})^{-\frac{1}{4}}$ $\hspace{3cm}$ (A4)

where $h_{tr0}$ is the reservoir levels at the beginning of the time step.

## A1.4 Routing

Given known flow directions, classically obtained from DEM, the cell to cell routing is done with a linear unit Gaussian hydrograph whose delay $\tau_i$ from node $i - 1$ to node $i$ is controlled by the routing velocity $v_i$ and the distance $d_i$ (see details in

Jay-Allemand et al. (2020)).





## A2 GR4

### A2.1 Production

The water balance is modeled with a production reservoir as described in section (A1.1) with equations A1 and A2, denoting the state $S$ and parameter $x_1$ instead of respectively $h_p$ and $c_p$.

### A2.2 Water exchange


A groundwater flow exchange term $F$ from the routing reservoir which depends on both the actual level in the store $R$, the reference level $x_3$ and a water exchange coefficient $x_2$ is taking into account both flow components

$$F = x_2 \left( \frac{R}{x_3} \right)^{\frac{7}{2}} \tag{A5}$$

### A2.3 Linear Routing


The 10% of the effective rainfall $P_r$ resulting from the excess of the production and the percolation is routed linearly using a unit hydrograph UH2 of time base $2x_4$, the remaining 90% is initially routed using UH1 of time base $x_4$. The ordinates of the UH are derived from their respective S hydrographs which also are functions of $x_4$

### A2.4 Non-linear routing

$$R = max(0; R + Q9 + F) \tag{A6}$$


$$Q_r = R \left\{ 1 - \left[ 1 + \left( \frac{R}{x_3} \right)^4 \right]^{-\frac{1}{4}} \right\} \tag{A7}$$

$$Q_d = max(0; Q1 + F) \tag{A8}$$

Total stream flow is given by

$$Q = Q_r + Q_d \tag{A9}$$

## A3 MARINE

### A3.1 Infitration


A Green and Ampt model is used and the infiltration $i(t)$ is described by equation A3.





### A3.2 Subsurface flow

The subsurface flow is based on the Darcy's law given by:

$$q(t) = T_o exp\left(\frac{\theta_s - \theta}{m}\right) tan\beta \tag{A10}$$

where $T_0$ is the local transmissivity of fully saturated soil $(m^2 s^{-1})$, $\theta_s$ and $\theta$ are saturated and local water contents $(m^3 m^{-3})$, $m$ is transmissivity decay parameter, and $\beta$ is local slope angle (rad).

### A3.3 Surface flow

The surface runoff is divided into overland flow and drainage flow, in both cases, the kinematic wave model is used assuming a 1-dimensional kinematic wave which is approximated with the Manning friction law. The equation is thus:

$$\frac{\partial h}{\partial t} + \frac{S_o^{0.5}}{n_0} \times \frac{5}{3} h^{\frac{2}{3}} \frac{\partial h}{\partial x} = r - i \tag{A11}$$

where $h$ is water depth (m), $t$ is time (s), $x$ is space variable (m), $r$ is rainfall rate $(ms^{-1})$, i is infiltration rate $(ms^{-1})$, $S_0$ stands for bed slope $(mm^{-1})$ and $n_o$ is the Manning friction parameter $(m^3/m^{-3})$ .

### Appendix B: Regionalized Sensitivity Analysis

Sensitivity analysis in hydrological modeling is defined as the investigation of the response function that links the variation
in the model outputs to changes in the input variables and/or parameters (e.g. review bySong et al. (2015)). It allows the determination of the relative contributions of different uncertainty sources to the variation in outputs using qualitative or quantitative approaches under a given set of assumptions and objectives.

A model can be sensitive to a parameter in two ways: a) the uncertainty in the parameter is propagated throughout the model thereby contributing in the overall model uncertainty. b) small change in the parameter results in significant change in the
output because of the high correlation between the output and the parameter.

A common method of global sensitivity analysis is the regionalized sensitivity analysis (RSA). This is also called generalized sensitivity analysis and it is based on Monte Carlo simulations. Parameter values are taken here from uniform distributions within chosen ranges and then Monte Carlo simulations are run using the parameter sets and based on a defined goodness of fit criteria (GOF), the results are then classified as behavioral or non-behavioral based on a chosen threshold of the GOF criteria.
For each parameter, the difference between the cumulative distribution of the behavioral and non-behavioral sets is determined using a quantitative Kolmogorov-Smirnov (KS) statistic 3. A significant difference means the parameter is sensitive, and the



larger the difference, the more sensitive the parameter is. This method has been widely used in hydrological modeling and is easy to implement (Song et al., 2015), however the choice of the GOF criteria and threshold is highly subjective.

$$KS(x_i) = \max_{y} \left| F_y(y) - F_{y|x_i}(y) \right| \tag{B1}$$

This approach to sensitivity is used in the course of the present study to investigate the sensitivity of the parameters of the three models on the study catchments.

## Appendix C:  Tables of events comparisons

**Table C1.** Performance evaluation efficiencies of the three models at discharge simulations on the **Gardon**

|  | NSE | | | KGE | | | r | | | beta | | | alpha | | |
|---|---|---|---|---|---|---|---|---|---|---|---|---|---|---|---|
| Event | GR4 | SMASH | Marine | GR4 | SMASH | Marine | GR4 | SMASH | Marine | GR4 | SMASH | Marine | GR4 | SMASH | Marine |
| Ev_10_11_2008 | 0.66 | **0.90** | 0.82 | 0.58 | **0.89** | 0.56 | 0.92 | 0.95 | **0.98** | 1.31 | **0.99** | 1.4 | 1.28 | **0.90** | 1.17 |
| Ev_01_11_2011 | 0.61 | **0.91** | 0.66 | 0.59 | **0.81** | 0.50 | 0.92 | **0.98** | 0.98 | 1.17 | **1.12** | 1.32 | 1.36 | **1.14** | 1.39 |
| Ev_16_09_2014 | 0.07 | **0.66** | 0.50 | -0.04 | 0.54 | **0.69** | 0.43 | **0.86** | 0.79 | 0.61 | 1.30 | **0.80** | 0.23 | 0.69 | **1.13** |
| Ev_09_10_2014 | 0.68 | **0.72** | 0.69 | 0.52 | **0.62** | 0.47 | 0.91 | 0.89 | **0.94** | 0.72 | **0.88** | 0.63 | 0.62 | **0.65** | 0.62 |
| Ev_10_09_2015 | 0.16 | 0.60 | **0.91** | 0 | 0.45 | **0.71** | 0.66 | 0.84 | **0.98** | 0.44 | 1.52 | **0.78** | 0.25 | **0.99** | 0.81 |
| Ev_27_10_2015 | 0.67 | 0.58 | **0.79** | 0.43 | 0.39 | **0.45** | 0.94 | 0.87 | **0.98** | 0.75 | **0.73** | 0.58 | 0.49 | 0.47 | **0.64** |
| Ev_22_11_2018 | **0.82** | 0.81 | 0.09 | **0.86** | 0.67 | -0.01 | 0.91 | 0.94 | **0.95** | **1.05** | 1.06 | 0.33 | **0.91** | 0.68 | 0.24 |
| Ev_08_11_2018 | **0.93** | 0.91 | 0.19 | **0.93** | 0.79 | 0.16 | 0.97 | **0.97** | 0.94 | 1.06 | **1.01** | 0.4 | **1.0** | 0.79 | 0.41 |
| **Average** | 0.58 | **0.76** | 0.58 | 0.48 | **0.65** | 0.44 | 0.83 | 0.91 | **0.94** | 0.89 | **1.08** | 0.78 | 0.77 | 0.79 | **0.80** |

*Author contributions.*  AH performed the numerical simulations and prepared the paper. PAG implemented the Green and Ampt operator and Monte Carlo algorithm to SMASH numerical assimilation platform. AH implemented the masked Monte Carlo algorithm. MJ-A imple-

mented the SMASH routing operator, the variational data assimilation algorithm along with the masked calibration method. HR provided MARINE model and physiographic data. PAG, HR, PJ surpervised the work. All authors participated to discussions, results analysis and paper writing.

*Competing interests.*  The authors declare that they have no competing interests





**Table C2.** Comparison of the SMASH, GR4 and MARINE in terms of some hydrological signatures; the Percentage Peak difference (PPD), the runoff coefficient (CR), the time difference of the peak (PD) and the Synchronous Percentage of the Peak Discharge (SSPD) on the **Gardon**

| Event | PPD | | | CR | | | | Peak Diff. (hr) | | | SSPD (%) | | |
|---|---|---|---|---|---|---|---|---|---|---|---|---|---|
| | GR4 | SMASH | Marine | obs | GR4 | SMASH | Marine | GR4 | SMASH | Marine | GR4 | SMASH | Marine |
| Ev_10_11_2008 | **114** | 76 | 117 | 0.37 | 0.48 | **0.36** | 0.52 | -2 | **0** | **0** | 88 | 76 | 117 |
| Ev_01_11_2011 | 120 | **110** | 149 | 0.50 | 0.59 | **0.56** | 0.67 | -3 | **0** | **0** | **102** | 110 | 149 |
| Ev_16_09_2014 | 15 | 57 | **141** | 0.31 | 0.19 | 0.41 | **0.25** | 58 | **1** | **-1** | 7 | 56 | **90** |
| Ev_09_10_2014 | 40 | 38 | **60** | 0.59 | 0.43 | **0.52** | 0.38 | 1 | **0** | **0** | 40 | 38 | **60** |
| Ev_10_09_2015 | 14 | 58 | **81** | 0.25 | 0.11 | 0.37 | **0.21** | 5 | 4 | **0** | 7 | 52 | **81** |
| Ev_27_10_2015 | 39 | 36 | **67** | 0.37 | **0.28** | 0.27 | 0.22 | **0** | **0** | -1 | 39 | 36 | **66** |
| Ev_22_11_2018 | **69** | 54 | 23 | 0.51 | **0.54** | **0.54** | 0.17 | -1 | **0** | **0** | 67 | 54 | 23 |
| Ev_08_11_2018 | **94** | 77 | 43 | 0.48 | 0.51 | **0.48** | 0.19 | -1 | **0** | **0** | 90 | 77 | 43 |
| **Average** | 63.1 | 63.3 | **85.1** | 0.42 | 0.39 | 0.44 | 0.33 | -7.1 | -0.63 | **0.25** | 55 | 62.4 | **78.6** |

**Table C3.** Performance evaluation efficiencies of the three models at discharge simulations on the **Ardeche**

| Event | NSE | | | KGE | | | r | | | beta | | | alpha | | |
|---|---|---|---|---|---|---|---|---|---|---|---|---|---|---|---|
| | GR4 | SMASH | Marine | GR4 | SMASH | Marine | GR4 | SMASH | Marine | GR4 | SMASH | Marine | GR4 | SMASH | Marine |
| Ev_2008_10_19 | 0.74 | **0.92** | 0.79 | 0.57 | **0.81** | 0.53 | 0.92 | **0.97** | 0.95 | 0.84 | **0.89** | 1.41 | 0.61 | **0.86** | 0.78 |
| Ev_2010_05_11 | 0.16 | **0.68** | 0.47 | 0.28 | 0.59 | **0.69** | 0.78 | **0.92** | 0.77 | 0.67 | **0.83** | 1.19 | 0.40 | 0.63 | **0.91** |
| Ev_2010_09_06 | 0.28 | 0.66 | **0.73** | 0.07 | **0.63** | 0.45 | 0.87 | 0.82 | **0.96** | 0.43 | **0.91** | 1.48 | 0.28 | 0.69 | **1.26** |
| Ev_2011_11_02 | 0.72 | **0.94** | 0.84 | 0.75 | **0.90** | 0.76 | 0.91 | **0.98** | **0.98** | **0.98** | 0.93 | 1.21 | 1.23 | **1.06** | 1.12 |
| Ev_2014_09_18 | 0.71 | 0.38 | **0.86** | **0.79** | 0.34 | 0.76 | 0.86 | 0.91 | **0.95** | 1.15 | 1.48 | **1.13** | **0.97** | 1.44 | 0.80 |
| Ev_2014_11_14 | 0.55 | 0.80 | **0.87** | 0.55 | **0.89** | 0.81 | 0.92 | 0.90 | **0.96** | 1.18 | 0.95 | **0.96** | 1.40 | **1.02** | 1.19 |
| Ev_2019_04_23 | 0.89 | **0.93** | 0.85 | 0.72 | **0.92** | 0.67 | **0.98** | 0.97 | 0.67 | 0.85 | **0.94** | 1.08 | 0.76 | **1.04** | 0.68 |
| **Average** | 0.58 | 0.76 | **0.77** | 0.53 | **0.73** | 0.67 | 0.89 | 0.92 | **0.96** | 0.87 | **0.99** | 1.13 | 0.81 | **0.96** | **0.96** |

*Acknowledgements.* The first author would like to acknowledge INRAe for funding the internship leading to this research and the support of
Petroleum Technology Development Fund (PTDF), Nigeria for funding the Master program.



**Table C4.** Comparison of the SMASH, GR4 and MARINE in terms of some hydrological signatures; the Percentage Peak difference (PPD), the runoff coefficient (CR), the time difference of the peak (PD); and the Synchronous Percentage of the Peak Discharge (SSPD) on **Ardeche**

| Event | PPD | | | CR | | | | Peak Diff. (hr) | | | SSPD (%) | | |
|---|---|---|---|---|---|---|---|---|---|---|---|---|---|
| | GR4 | SMASH | Marine | obs | GR4 | SMASH | Marine | GR4 | SMASH | Marine | GR4 | SMASH | Marine |
| Ev_2008_10_19 | 54 | 78 | **93** | 0.60 | 0.52 | **0.55** | 0.85 | 3 | **1** | **1** | 46 | 74 | **89** |
| Ev_2010_05_11 | 45 | 57 | **71** | 0.50 | 0.34 | **0.42** | 0.59 | -1 | 1 | **0** | 44 | 56 | **71** |
| Ev_2010_09_06 | 20 | 47 | **105** | 0.21 | 0.09 | 0.19 | **0.31** | 1 | 1 | **1** | 19 | 42 | **94** |
| Ev_2011_11_02 | 115 | **92** | 129 | 0.73 | **0.73** | 0.69 | 0.89 | **-1** | 3 | **-1** | 115 | 86 | 127 |
| Ev_2014_09_18 | 80 | **115** | 70 | 0.43 | **0.51** | 0.66 | 0.53 | **2** | **2** | -2 | 66 | **106** | 63 |
| Ev_2014_11_14 | **110** | 75 | 116 | 0.79 | 0.94 | **0.76** | **0.76** | -1 | 1 | **0** | **108** | 71 | 116 |
| Ev_2019_04_23 | 72 | **94** | 86 | 0.61 | 0.53 | **0.59** | 0.66 | 2 | 2 | **0** | 68 | 84 | **86** |
| **Average** | 70.9 | 79.7 | **95.7** | 0.55 | 0.52 | **0.55** | 0.65 | -0.71 | -1.57 | **0.14** | 66.6 | 74 | **92.3** |

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
