# Peer review of "Signature and sensitivity-based comparison of conceptual and process oriented models, GR4H, MARINE and SMASH, on French Mediterranean flash floods"

_Hydrology and Earth System Sciences, 2021_

## Author Comment (AC1)

**Reviewer Comments**

February 4, 2022

Dear reviewer,
We thank you very much for this detailed and constructive review that will help us to clarify and sharpen our article. We agree to seriously address the major and minor comments you raised with substantial rework especially on the following points:

- Improvement of the bibliographical synthesis regarding hydrological models inter-comparison, based on references you provided.

- Clarification of the outline and goals with a more concise story line, and clarification of the proposed methodology with three levels of analysis.

- Making a more straight to the point presentation of the results, as well as deepening their analysis.

- Substantial reorganization of the paper, which includes:

    Using a table to summarize models features, and another table to summarize basin characteristics.

    Moving sensitivity analysis (SA) results to the appendix and summarizing them in the main text in relation with other results on model inter-comparison and signature analysis.

    Better justifying some methodological choices (SA setup and use of SIM outputs in comparison of simulated states).

- Improving discussion and more precise conclusions on "how and why" the models perform differently, more general insights, and better connections with study goals and research questions.

We give in the following, some elements of answer to your main questions on methodological choices on: calibration routines, sensitivity analysis temporal windows, and use of soil states simulated by the SIM model.

- Concerning the question related to considering three models with their own calibration routine. Each of the three structurally different hydrological models are effectively calibrated using their respective optimization algorithms that are adapted to the complexity of of each model's parameterization. This enables efficient global optimizations of parameter sets for each model, with classical cost functions evaluating the misfit between simulated and observed discharges in a global manner in time (either on relatively long time series for the two continuous models or on several events at the same time for the event based model). This point will be better explained and discussed.

- The global sensitivity analysis is in line with the models' calibration methods, and aims to assess the global sensitivity of each model to its parameters, on the same setup as that of calibration. This part will be reworked, including reorganization, improved analysis and links to other analysis and general discussions/conclusions.

- Regarding the use of SIM model in the comparison of the simulated states. This choice is motivated by the fact that SIM is a well validated surface model with a rich description of soil atmosphere processes, applied on a wide spatio-temporal domain which ensures good data availability. Moreover, there is perhaps a misunderstanding, SIM runs at relatively fine time steps but we only used daily quantities for sake of simplicity. Moreover, SIM1 is traditionally used to initialize MARINE, hence the choice is made to use SIM2 as benchmark, whose parameterization is more complex (ex. more soil levels, which is not usable for MARINE

initialization). Note that using satellite moisture data is a work on its own and is not within the scope of this study; it was already studied with MARINE in Eeckman (2020). These points will be clarified and better discussed.

We will be happy to revise our paper following your helpful comments and also to provide a detailed response letter.